# 🕵️ Evidence-Guided Multi-Image Reasoning in Visual Retrieval-Augmented Generation

## Abstract

Visual retrieval-augmented generation (VRAG) augments vision–language models (VLMs) with external visual knowledge to ground reasoning and reduce hallucinations. Yet current VRAG systems often fail to reliably perceive and integrate evidence across multiple images, leading to weak grounding and erroneous conclusions. In this paper, we propose EVisRAG, an end-to-end framework that learns to reason with evidence-guided multi-image to address this issue. The model first observes retrieved images and records per-image evidence, then derives the final answer from the aggregated evidence. To train EVisRAG effectively, we introduce Reward-Scoped Group Relative Policy Optimization (RS-GRPO), which binds fine-grained rewards to scope-specific tokens to jointly optimize visual perception and reasoning abilities of VLMs. Experimental results on multiple visual question answering benchmarks demonstrate that EVisRAG delivers substantial end-to-end gains over backbone VLM with 27% improvements on average. Further analysis shows that, powered by RS-GRPO, EVisRAG improves answer accuracy by precisely perceiving and localizing question-relevant evidence across multiple images and deriving the final answer from that evidence, much like a real detective.

## 1 Introduction

Retrieval-Augmented Generation (RAG) equips Large Language Models (LLMs) with a knowledge retriever that accesses a curated external knowledge base, supplying task-relevant context at generation time and mitigating hallucinations arising from insufficient parametric knowledge (Lewis et al., 2020; Asai et al., 2024). However, ineffective use of retrieved information limits practical adoption in domain-specific tasks. Retrieval-augmented reasoning addresses this gap by extracting evidence from external knowledge during the reasoning process. When combined with reinforcement learning (RL) optimization (Shao et al., 2024; Rafailov et al., 2023; Schulman et al., 2017), this paradigm improves the model's ability to leverage retrieved evidence and tackle higher-difficulty queries (Li et al., 2025; Song et al., 2025). Yet a substantial portion of real-world knowledge exists in non-textual modalities, such as images, tables, and complex document layouts. Preprocessing routes that first linearize these signals via image captioning or OCR and then feed only text to LLMs inevitably discard crucial visual and spatial cues, preventing the model from accessing information originally present in images or document pages (Zhang et al., 2024b).

To address this limitation, Visual RAG (VRAG) (Yu et al., 2025; Faysse et al., 2024a) retrieves document page snapshots as units, preserving visual and spatial cues so VLMs can read evidence directly from images. Recent variants couple retrieval with reinforcement learning, inserting retrieved images into intermediate reasoning steps so the model can derive the correct answer from pixels rather than text alone (Peng et al., 2025; Wu et al., 2025). Despite these gains, many methods still transplant text-based RAG practices to vision and ignore modality-specific needs such as cross-image grounding, layout-aware reading, and region-level attention. As a result, models often fail to perceive information reliably across multiple images. Some works introduce perception-oriented actions or auxiliary agents to guide reasoning (Wang et al., 2025b;a), which improves attention to visual detail but increases architectural complexity and computational cost, complicating end-to-end training and later reconfiguration.

Recent advances in vision-language reasoning models (VLRMs) have introduced promising strategies for enhancing visual perception on a single image during the reasoning process (Shen et al.,

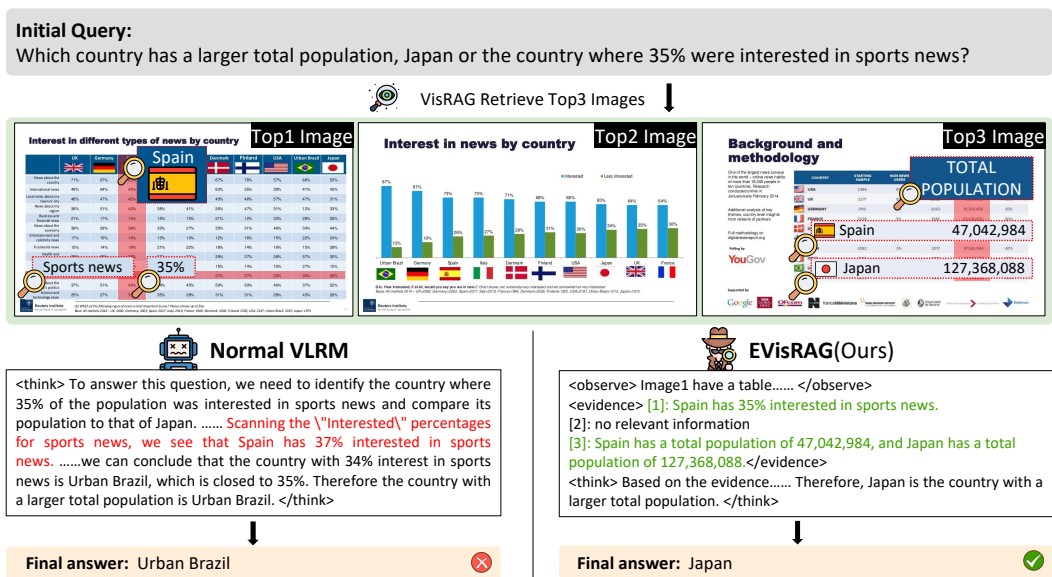

Figure 1: Comparison of normal vision-language reasoning model (VLRM) and EVisRAG

2025; Xu et al., 2025) by incorporating auxiliary rewards related to visual perception. Although these VLRMs perform well on single-image inputs, VRAG often retrieves multiple images, requiring cross-image localization and integration of fine-grained evidence. While current methods lack a built-in per-image evidence collection and instead rely on external tools or agents, increasing complexity and instability. In addition, current VLRM training strategies typically optimize perception and reasoning with mixed rewards, overlooking the effective scope and objective differences of each signal, which blurs credit assignment and causes interference.

Motivated by these challenges, we propose **E**vidence-guided **Vis**ion **R**etrieval-**a**ugmented **G**eneration (EVisRAG) to equip VLMs with precise visual perception in multi-image scenarios. As illustrated in Figure 1, EVisRAG conducts a linguistic observation phase that sequentially gathers evidence from retrieved images, maintaining focus on them, and then performs reasoning on the collected evidence to derive the correct answer. To train EVisRAG effectively, we introduce Reward-Scoped Group Relative Policy Optimization (RS-GRPO), a method that uses fine-grained rewards applied directly to in-scope tokens to jointly optimize visual perception and reasoning. Experiments on different VQA tasks demonstrate the effectiveness of EVisRAG, showing substantial improvements over different VRAGs. Powered by RS-GRPO, EVisRAG can precisely find question-relevant evidence image by image and then reason over the recorded cues to produce grounded answers just like a detective. Moreover, EVisRAG demonstrates stronger visual perception and higher answer accuracy among other baselines, confirming that richer visual perception improves the ability of question understanding and response quality.

## 2 RELATED WORK

Early research on retrieval-augmented generation (RAG) equips large language models (LLMs) with retrievers over curated corpora to provide task-relevant context and mitigate hallucinations (Lewis et al., 2020; Asai et al., 2024). Building on this foundation, retrieval-augmented reasoning acquires evidence at intermediate steps and uses it to guide reasoning (Shao et al., 2024; Rafailov et al., 2023; Schulman et al., 2017; Li et al., 2025; Song et al., 2025). Nevertheless, a considerable portion of real-world knowledge is non-textual, residing in images, tables, and documents with complex layouts. Pipelines that first linearize these signals through captioning or optical character recognition and then supply only text to the model often discard essential visual and spatial cues, which reduces reliability on downstream tasks (Zhang et al., 2024b).

To address the problem, VisRAG (Yu et al., 2025) and Colpali (Faysse et al., 2024a) introduce Visual RAG (VRAG), which uses document page snapshots as retrieval units, enabling vision-language

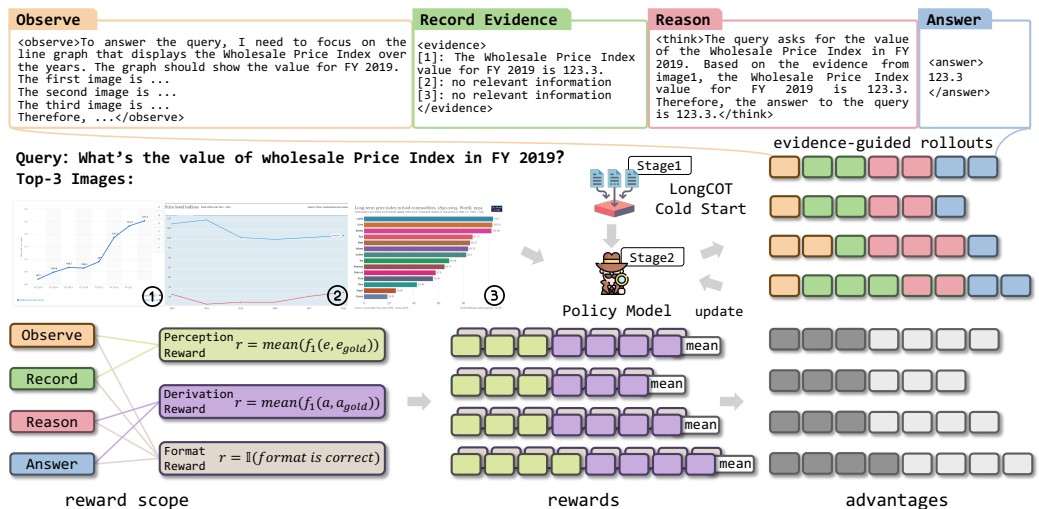

Figure 2: Overall framework of EVisRAG. Followed by the query and top-3 retrieved document pages, EVisRAG outputs four token scopes: observe, record evidence, reason, and answer. RS-GRPO assigns three fine-grained rewards to scope-specific tokens. In-scope rewards are then averaged and group-normalized to obtain token advantages for policy updates.

models to read evidence directly from images. Recent studies further couple retrieval with reinforcement learning: R1-Router (Peng et al., 2025) and MMSearch-R1 (Wu et al., 2025) allow the model to decide when and where to retrieve and insert retrieved images into intermediate reasoning steps, so that answers are derived from visual content rather than text alone. Despite this progress, many approaches transplant text-centric RAG practices to the visual modality and insufficiently address modality-specific needs, such as cross-image grounding, layout-aware reading, and region-level attention, which leads to unstable perception across multiple images. Several studies have recognized this limitation: VRAG-RL (Wang et al., 2025b) defines a visual perception action space that covers region selection, cropping, and scaling, allowing the model to revisit image content multiple times during reasoning; ViDoRAG (Wang et al., 2025a) introduces a multi-agent framework that decouples perception from reasoning, allowing the model to focus on a single subtask at each step. While these frameworks are effective, they increase architectural complexity and computational overhead, complicating end-to-end training and subsequent reconfiguration.

Recent advances in vision and language reasoning models (VLRMs) have introduced effective strategies for strengthening visual perception during reasoning. Vision-R1 (Zhan et al., 2025), MM-Eureka (Meng et al., 2025), Ocean-R1 (Lingfeng et al., 2025), ThinkLite-VL (Wang et al., 2025c), and OpenVLThinker (Deng et al., 2025) show that directly applying GRPO, sometimes even without supervised fine-tuning, substantially promotes the emergence of chain of thought reasoning and can elicit "aha" moments. VLM-R1 (Shen et al., 2025) and Mixed-R1 (Xu et al., 2025) further improve perceptual grounding by augmenting answer correctness signals with auxiliary perception rewards, encouraging better use of image information. However, in VRAG settings that require reasoning over semantically rich content from multiple images, the remaining limitations in perceptual grounding often lead to misinterpretation of visual evidence, which in turn undermines the validity of the overall reasoning process.

# 3 METHODOLOGY

This section introduces our method, EVisRAG, which enables VLMs to reason over multiple images with rich visual evidence. We first provide an overview of the evidence-guided reasoning process of EVisRAG, covering observation, evidence recording, and answer reasoning (Section 3.1). We then describe how EVisRAG strengthens fine-grained perceptual grounding during reasoning through the Reward-Scoped Group Relative Policy Optimization(RS-GRPO) algorithm(Section 3.2).

### 3.1 THE OVERVIEW FRAMEWORK OF EVISRAG

Given a query $q$ and corpus $\mathcal{D}$ of document-page snapshots, EVisRAG performs a step-by-step reasoning to retrieve and localize visual evidence and produce the final answer $a$:

$$(q, \mathcal{D}) \xrightarrow{\text{EVisRAG}} a, \tag{1}$$

where $q$ is an open-domain question and corpus $D$ indexes page-level images, providing visual evidence relevant to $q$ for a VLM to exploit.

**Information Retrieving.** The first stage of EVisRAG aims to retrieve a set of pages from a large document page corpus $\mathcal{D}$, given a query $q$. Following the VisRAG (Yu et al., 2025) retrieval paradigm, we obtain

$$\mathcal{D}_\mathcal{R} = \text{VisRAG-Ret}(q, \mathcal{D}), \tag{2}$$

where the candidate set $\mathcal{D}_\mathcal{R} \subset \mathcal{D}$ contains the top-$k$ document pages relevant to the question $q$.

**Visual Perception.** After gathering candidates $\mathcal{D}_R = \{d_i\}_{i=1}^k$ from $\mathcal{D}$, EVisRAG sequentially observes these pages and produces a coarse, page-aware description $r_{\text{observe}}$:

$$P(r_{\text{observe}} \mid q, \mathcal{D}_R) = \prod_{t=1}^{|\mathcal{T}_o|} P(r_{\text{observe},t} \mid r_{\text{observe},<t}, q, \mathcal{D}_R), \tag{3}$$

where $|\mathcal{T}_o|$ is the length of the observation sequence and $r_{\text{observe},t}$ denotes its $t$-th token.

Conditioned on $q$ and $r_{\text{observe}}$, EVisRAG then records evidence from each page by generating per-image evidence sequences $r_{\text{evidence}}^{(i)}$:

$$P(r_{\text{evidence}} \mid q, r_{\text{observe}}, \mathcal{D}_R) = \prod_{i=1}^{k} \prod_{t=1}^{|\mathcal{T}_e^{(i)}|} P\left(r_{\text{evidence},t}^{(i)} \mid r_{\text{evidence},<t}^{(i)}, q, r_{\text{observe}}, d_i\right), \tag{4}$$

where $r_{\text{evidence}} = \{r_{\text{evidence}}^{(i)}\}_{i=1}^k$, $|\mathcal{T}_e^{(i)}|$ is the length of the evidence sequence for the retrieved document page $d_i$, and $r_{\text{evidence},t}^{(i)}$ is its $t$-th token.

**Answer Reasoning.** After visual perception, EVisRAG conducts detective-style reasoning over the perceived information $r_{\text{perception}} = \{r_{\text{observe}}, r_{\text{evidence}}\}$: it distills leads from the recorded evidence, formulates and tests hypotheses across pages, cross-checks contradictions, and organizes a coherent reasoning trajectory $r_{\text{reason}}$:

$$P(r_{\text{reason}} \mid q, r_{\text{perception}}, \mathcal{D}_R) = \prod_{t=1}^{|\mathcal{T}_r|} P(r_{\text{reason},t} \mid r_{\text{reason},<t}, q, r_{\text{perception}}, \mathcal{D}_R), \tag{5}$$

where $|\mathcal{T}_r|$ is the length of the reasoning sequence and $r_{\text{reason},t}$ denotes its $t$-th token.

Conditioned on $q$, $r_{\text{perception}}$, and $r_{\text{reason}}$, EVisRAG then produces the final answer sequence $r_{\text{answer}}$:

$$P(r_{\text{answer}} \mid q, r_{\text{perception}}, r_{\text{reason}}, \mathcal{D}_R) = \prod_{t=1}^{|\mathcal{T}_a|} P(r_{\text{answer},t} \mid r_{\text{answer},<t}, q, r_{\text{perception}}, r_{\text{reason}}, \mathcal{D}_R), \tag{6}$$

where $|\mathcal{T}_a|$ is the answer length and $r_{\text{answer},t}$ is its $t$-th token.

### 3.2 OPTIMIZING VLMS TO EVIDENCE-GUIDED REASON USING RS-GRPO

To enhance EVisRAG's ability to accurately record evidence from multiple images and reason based on that evidence, we employ a two-stage training as shown in Figure 2. In the first stage, we apply supervised fine-tuning (SFT) as a cold start. In the second stage, we introduce Reward-Scoped Group Relative Policy Optimization (RS-GRPO), which extends GRPO to jointly optimize perception and reasoning ability of VLMs, with fine-grained rewards applied to their corresponding reward scopes.

**Reward Scopes.** To evaluate model outputs while encouraging the evidence-guided reasoning paradigm, we adopt three fine-grained rewards in a coordinated scheme. The format reward $R_{\text{format}}$

enforces adherence to an evidence-guided reasoning paradigm by requiring the model to observe, record evidence, reason, and answer in a disciplined order, making intermediate steps explicit and supervision stable. The perception reward $R_{\text{perception}}$ checks whether question-relevant regions are correctly localized and summarized for each image based on the ground truth evidence generated by a larger VLM and allows an explicit no relevant information when evidence is absent. The derivation reward $R_{\text{derivation}}$ evaluates whether the model derives the correct final answer from its visual perception, ensuring the reasoning is grounded in the observed and recorded evidence. More details of the reward design are shown in Appendix A.3

To jointly train the perception and reasoning ability of VLMs, we introduce Reward Scopes, which route supervision to scope-specific tokens to sharpen credit assignment, reduce interference, and stabilize training. Let $\mathcal{M}(t)$ denote the set of reward channels applicable to the token at position $t$. The output sequence is segmented by special tokens into four scopes, the observe scope $\mathcal{T}_o$, the record evidence scope $\mathcal{T}_e$, the reason scope $\mathcal{T}_r$, and the answer scope $\mathcal{T}_a$. Rewards act only where they are meaningful. $R_{\text{perception}}$ supervises tokens in $\mathcal{T}_o$ and $\mathcal{T}_e$, guiding the model to summarize the right visual regions. $R_{\text{derivation}}$ supervises tokens in $\mathcal{T}_r$ and $\mathcal{T}_a$, encouraging the model to derive the correct final answer from what was perceived. $R_{\text{format}}$ applies to all tokens and keeps the evidence-guided workflow explicit and stable. Formally, we define the reward–scope mapping as:

$$\mathcal{M}(t) = \begin{cases} \{\, R_{\text{perception}},\ R_{\text{format}} \,\} & t \in \mathcal{T}_o \cup \mathcal{T}_e \\ \{\, R_{\text{derivation}},\ R_{\text{format}} \,\} & t \in \mathcal{T}_r \cup \mathcal{T}_a \end{cases}. \tag{7}$$

For the $i$-th sampled output and its token at position $t$, let $R_t^{(m),i}$ denote the score from reward channel $m \in \mathcal{M}(t)$. The scope-aggregated token reward is the mean over its in-scope channels:

$$\bar{R}_t^i = \frac{1}{|\mathcal{M}(t)|} \sum_{m \in \mathcal{M}(t)} R_t^{(m),i}. \tag{8}$$

**RS-GRPO objective.** To train both visual perception and reasoning, EVisRAG adopts an RS-GRPO objective that explicitly computes token advantages under reward scopes. Given a group of $G$ sampled outputs, the token-level advantage is

$$\hat{A}_t^i \;=\; \frac{\bar{R}_t^i \;-\; \text{mean}\big(\{\bar{R}_t^1, \bar{R}_t^2, \ldots, \bar{R}_t^G\}\big)}{\text{std}\big(\{\bar{R}_t^1, \bar{R}_t^2, \ldots, \bar{R}_t^G\}\big)}, \tag{9}$$

where $i$ indexes the $i$-th sample in the group, and $G$ is the group size. We incorporate the resulting token-level advantages into DAPO to enhance exploration diversity and training stability, and optimize the model by minimizing the following objective:

$$\mathcal{L}_{\text{RS-GRPO}}(\theta) = -\frac{1}{\sum_{i=1}^{G} |o^i|} \sum_{i=1}^{G} \sum_{t=1}^{|o^i|} \min\Big( r_t^i(\theta)\, \hat{A}_t^i,\ \text{clip}\big(r_t^i(\theta),\, 1 - \epsilon_{\text{low}},\, 1 + \epsilon_{\text{high}}\big)\, \hat{A}_t^i \Big), \quad (10)$$

where $o^i$ is the $i$-th sampled output sequence, $r_t^i(\theta)$ is the importance ratio, and $\epsilon_{\text{low}}, \epsilon_{\text{high}}$ are the lower and upper clipping thresholds.

# 4 EXPERIMENTAL METHODOLOGY

This section describes the datasets, baselines, evaluation metrics, and implementation details.

**Datasets.** We first introduce the datasets used in our experiments, followed by the data statistics for golden reasoning trajectory construction.

We evaluate our EVisRAG on five visual question answering (VQA) tasks encompassing diverse document types, including ChartQA (Masry et al., 2022) and InfographicsVQA (Mathew et al., 2022) for various types of figures, MP-DocVQA (Tito et al., 2023) for industrial documents, Slide-VQA (Tanaka et al., 2023) for presentation slides, and ViDoseek (Wang et al., 2025a) for multi-document scenarios. For each query, we utilize VisRAG-Ret (Yu et al., 2025) to retrieve the top-3 relevant images as context. Subsequently, each question is categorized according to whether the

retrieved context provides sufficient information to answer the question with sufficient context or with insufficient context. More details of the test datasets are provided in Appendix A.1.

We collect 30,000 samples from the training sets of ChartQA and InfoVQA and divide them into SFT and GRPO subsets with an 8:2 split. For each query, VisRAG-Ret retrieves the top five images, while only the top three are used during testing. To build high-quality reasoning trajectories for EVisRAG, we employ Qwen2.5-VL (Bai et al., 2025) models to generate candidate chains of thoughts (Wei et al., 2022) and retain those yielding correct answers. Following (An et al., 2025), we filter out the data that can be easily answered when constructing the RL training data. This results in 60,000 SFT training samples and 4,000 RL training samples. More details of the data construction process are provided in Appendix A.2.

**Baselines.** All baselines use VisRAG-Ret for retrieval. For each query, we fetch the top-$k$ documents, then the model answers using the retrieved images and the original question.

We compare four groups. General VLMs include Qwen2.5-VL-7B, Qwen2.5-VL-32B (Bai et al., 2025) and MiMo-VL-7B-RL (Xiaomi, 2025). Two generation approaches form VisRAG-Gen(Yu et al., 2025). VLRMs trained on Qwen2.5-VL-7B-Instruct include Vision-R1-7B (Zhan et al., 2025), MM-Eureka-7B (Meng et al., 2025), Ocean-R1-7B (Lingfeng et al., 2025), ThinkLite-VL-7B (Wang et al., 2025c) and OpenVLThinker-7B (Deng et al., 2025). VRAG methods with the same backbone include R1-Router (Peng et al., 2025), MMSearch-R1 (Wu et al., 2025), and VRAG-RL (Wang et al., 2025b). More implementation details of the baseline methods are provided in Appendix A.6

**Evaluation Metrics.** Due to inherent limitations in retrieval, the selected context may or may not provide sufficient information to answer the query. To rigorously assess both the perceptual and reasoning capabilities of the model while mitigating the confounding effects of hallucination, we categorize each query-context pair into two types: sufficient context and insufficient context (Joren et al., 2025).

For queries where the retrieved images provide sufficient evidence, we adopt the original reference answer as the ground truth. When the context is inadequate to support a correct answer, the model is required to output "insufficient to answer." To evaluate overall performance under realistic VRAG settings, we report global *Accuracy* and *F1 Score* over all queries as comprehensive, dataset-level metrics.

Additional implementation details for the baseline methods are provided in Appendix A.6. We also compare three CoT approaches with our Evidence-guided prompt approach. Since these methods were not trained, we present them separately in Appendix A.4.

**Implementation Details.** We use Qwen2.5-VL-7B (Bai et al., 2025) as the backbone for our proposed EVisRAG. We use LLaMA-Factory (Zheng et al., 2024) and Easy-R1 (Yaowei et al., 2025) for open-sourcing the training framework that we used for SFT and GRPO. All experiments were executed on GPU clusters with computational capabilities comparable to NVIDIA A100 80GB GPUs. Further details on the hyperparameters that we used for SFT, GRPO are provided in Appendix A.5.

## 5 RESULTS AND ANALYSIS

In this section, we begin by evaluating the overall performance of EVisRAG on a range of VQA benchmarks, covering three single-hop and two multi-hop datasets. We subsequently perform ablation experiments to assess the contributions of our framework. Following this, we investigate how EVisRAG improves answer accuracy with visual perception.

### 5.1 OVERALL PERFORMANCE

Table 1 reports the overall results for EVisRAG and all baselines. EVisRAG-7B consistently outperforms every comparator across all benchmarks, with substantial gains over the Qwen2.5-VL-7B backbone, averaging +19% in accuracy and +27% in F1 score. These improvements indicate that an evidence-guided reasoning paradigm, coupled with RS-GRPO, strengthens perceptual grounding and enables reasoning that is explicitly conditioned on grounded evidence. Compared with RL-trained VLRMs, EVisRAG's explicit visual perception yields a clear advantage. Within the VLRM group, models emphasizing logical reasoning (e.g., OpenVLThinker (Deng et al., 2025)) do im-

Table 1: Overall Performance of EVisRAG and Baselines. **"Bold"** denotes the highest value. Meanwhile, the symbol "↑" indicates the increase in our highest value compared to the Vanilla baseline.

| Methods | In Distribution | | | | Out of Distribution | | | | | | Average | |
| --- | --- | --- | --- | --- | --- | --- | --- | --- | --- | --- | --- | --- |
| | ChartQA | | InfoVQA | | DocVQA | | SlideVQA | | ViDoSeek | | | |
| | Acc | F1 | Acc | F1 | Acc | F1 | Acc | F1 | Acc | F1 | Acc | F1 |
| **General VLMs** | | | | | | | | | | | | |
| Qwen2.5-VL-7B | 59.20 | 52.80 | 60.86 | 54.61 | 63.28 | 56.03 | 51.62 | 46.11 | 42.56 | 42.48 | 55.50 | 50.41 |
| MiMo-VL-7B-RL | 54.96 | 40.59 | 68.11 | 45.93 | 74.11 | 47.67 | 77.88 | 47.45 | 48.34 | 38.30 | 64.68 | 43.99 |
| Qwen2.5-VL-32B | 69.12 | 60.58 | 78.13 | 66.06 | 83.93 | 73.78 | 78.42 | 58.65 | 47.55 | 52.78 | 71.43 | 62.37 |
| **VisRAG-Gen** | | | | | | | | | | | | |
| Page Concatenation | 59.20 | 52.80 | 52.92 | 46.42 | 60.58 | 47.84 | 64.57 | 48.45 | 45.01 | 41.37 | 56.63 | 47.63 |
| Weighted Selection | 32.24 | 32.32 | 25.07 | 27.36 | 33.67 | 37.11 | 33.81 | 36.44 | 21.98 | 31.64 | 29.35 | 32.97 |
| **VLRMs** | | | | | | | | | | | | |
| Vision-R1 | 56.16 | 50.84 | 29.53 | 27.73 | 32.49 | 30.04 | 52.34 | 47.51 | 39.05 | 37.82 | 41.91 | 38.79 |
| Ocean-R1-7B | 47.68 | 47.58 | 53.20 | 53.49 | 56.35 | 57.02 | 60.07 | 57.86 | 40.63 | 46.75 | 51.59 | 52.54 |
| MM-Eureka | 64.32 | 58.28 | 40.53 | 40.32 | 56.68 | 54.75 | 63.49 | 58.68 | 44.40 | 47.25 | 53.88 | 51.86 |
| ThinkLite-VL-7B | 57.60 | 53.53 | 61.70 | 61.62 | 62.61 | 62.37 | 65.29 | 63.30 | 45.18 | 48.40 | 58.48 | 57.84 |
| OpenVLThinker | 67.60 | 62.72 | 70.47 | 70.51 | 71.74 | 72.51 | 73.02 | 72.63 | 43.52 | 57.27 | 65.27 | 67.13 |
| **VRAGs** | | | | | | | | | | | | |
| MMSearch-R1 | 63.28 | 59.89 | 57.94 | 57.71 | 61.59 | 60.82 | 65.29 | 60.97 | 44.40 | 54.34 | 58.50 | 58.75 |
| VRAG-RL | 47.00 | 10.03 | 64.86 | 12.21 | 73.22 | 22.39 | 73.85 | 15.37 | 43.82 | 18.14 | 60.55 | 15.63 |
| R1-Router | 60.72 | 15.53 | 60.58 | 15.17 | 75.97 | 25.25 | 75.36 | 17.21 | 44.66 | 12.53 | 63.46 | 17.14 |
| **EVisRAG(ours)** | | | | | | | | | | | | |
| **EVisRAG-3B** | 72.64 | 72.54 | 71.03 | 71.83 | 78.17 | 79.30 | 75.84 | 75.49 | 45.71 | 60.13 | 68.68 | 71.86 |
| **EVisRAG-7B** | **76.80** | **76.60** | **79.39** | **79.80** | **85.45** | **86.82** | **81.29** | **80.28** | **52.10** | **65.78** | **75.01** | **77.86** |
| Δours → Qwen7B | 17.60↑ | 23.80↑ | 18.53↑ | 25.19↑ | 22.17↑ | 30.79↑ | 29.67↑ | 34.17↑ | 9.54↑ | 23.30↑ | 19.51↑ | 27.45↑ |
| Δours → OpenVLThinker | 9.20↑ | 13.88↑ | 8.92↑ | 9.29↑ | 13.71↑ | 14.31↑ | 8.27↑ | 7.65↑ | 8.58↑ | 8.51↑ | 9.74↑ | 10.73↑ |

prove question answering performance, underscoring the value of stronger reasoning. Nevertheless, EVisRAG's added perceptual grounding closes a further gap. Moreover, the three VRAG models improve the extraction of key evidence from retrieved context and, owing to their generalization, outperform the backbone when reasoning over multiple images. Yet they remain more than ten percentage points below EVisRAG-7B, since they neglect the need for strong perceptual grounding over multiple images with rich visual information. EVisRAG further allows a 7B parameter model to exceed the performance of considerably larger 32B parameter models. Furthermore, thrived on our RS-GRPO algorithm, EVisRAG jointly improves both perception and reasoning capabilities of VLMs, leading to a more effective and adaptable RAG framework.

## 5.2 ABLATION STUDY

This section reports ablation studies that isolate three training strategies, including the evidence-guided reasoning paradigm, perception reward, and Reward-Scoped Group Relative Policy Optimization (RS-GRPO), to assess the effectiveness of EVisRAG.

As shown in Table 2, EVisRAG achieves the best results across datasets, demonstrating that the evidence-guided reasoning paradigm combined with RS-GRPO enables VLMs to precisely localize question-relevant evidence in each image and reason over the recorded cues to produce grounded answers. Training under a think-then-answer paradigm (w/o Perception) on the same data yields only modest gains, reflecting the absence of explicit mechanisms for perceptual grounding. Introducing evidence-guided reasoning paradigm while rewarding only the final answer (w/o Perception Reward) leads to additional improvements, while underscores EVisRAG demonstrating that introducing perception reward can further enhance the perception ability of VLMs. Augmenting GRPO with a perception reward but without reward-scoped (w/o RS-GRPO) provides a further increment, yet the uniformly aggregated rewards dilute guidance across tokens. In contrast, RS-GRPO applies rewards directly within their designated reward scopes, sharpening credit assignment, stabilizing optimization, and ultimately delivering the strongest overall performance.

Table 2: Ablation study on accuracy (%) averaged over 5 runs with different random seeds. We report mean ± standard deviation: "w/o Perception" trains the model with a standard think-then-answer approach on the same data. "w/o Perception Reward" uses only answer correctness as the reward, omitting the additional Perception Reward. "w/o RS-GRPO" sums all rewards and applies them to every token, corresponding to the standard GRPO algorithm.

| Methods | In Distribution | | Out of Distribution | | | Avg. Acc |
|---------|-----------------|-----|---------------------|-----|-----|----------|
| | ChartQA | InfoVQA | DocVQA | SlideVQA | ViDoSeek | |
| EVisRAG (Ours) | **76.8** ± 0.6 | **79.2** ± 0.7 | **85.5** ± 1.2 | **81.3** ± 1.0 | **51.8** ± 0.7 | **74.9** ± 0.8 |
| w/o Perception | 67.2 ± 1.3 | 73.3 ± 1.6 | 75.7 ± 2.1 | 77.3 ± 2.0 | 41.8 ± 1.2 | 67.1 ± 1.6 |
| w/o Perception Reward | 69.8 ± 1.1 | 74.2 ± 1.2 | 79.9 ± 3.5 | 77.5 ± 2.2 | 48.1 ± 1.7 | 69.9 ± 1.9 |
| w/o RS-GRPO | 72.0 ± 2.2 | 75.7 ± 2.1 | 80.0 ± 2.2 | 77.9 ± 1.7 | 48.7 ± 1.9 | 70.9 ± 2.0 |

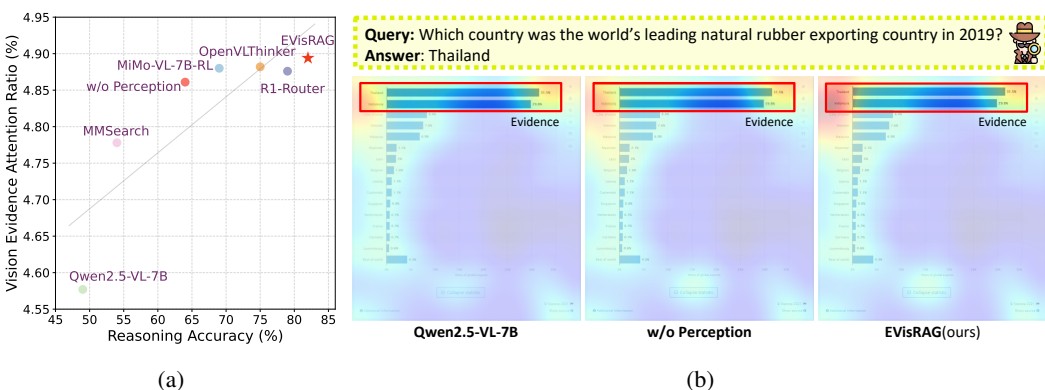

(a)  (b)

Figure 3: Comparison of models' attention to question-relevant visual evidence. (a) Accuracy vs. attention ratio within human-annotated boxes; EVisRAG achieves the highest. (b) Qualitative maps: Compared with the baseline, EVisRAG better focuses on the top bar encoding the evidence.

### 5.3 EVALUATING PERCEPTUAL ABILITY THROUGH VISUAL ATTENTION

In this section, we evaluate EVisRAG's visual perception using qualitative and quantitative evidence. Figure 3b shows a representative case for the question "Which country was the world's leading natural rubber exporting country in 2019?". Training the backbone under a think-then-answer paradigm (w/o Perception) modestly improves attention of VLMs to the legend and lower caption, helping the model read that the bars indicate shares of global exports. EVisRAG, with evidence-guided reasoning, further concentrates attention on the top bar region and correctly identifies Thailand as the leading exporter.

To quantify perception, we manually annotate evidence regions in more than 100 cases and compute the visual evidence attention ratio, defined as the percentage of attention mass falling inside the annotated evidence box. As shown in Figure 3a, EVisRAG achieves the highest reasoning accuracy and the highest visual evidence attention ratio among all baselines. The scatter also reveals a clear positive trend: higher attention to the evidence region is associated with higher answer accuracy.

### 5.4 VISUAL EVIDENCE DENSITY COMPARISON

We perform a visual evidence density analysis to evaluate the robustness of our method under varying numbers of images and different evidence densities. As shown in Figure 4, for each question, we retrieve the top-1 to top-5 images as context, which we refer to as references. Within these references, image tokens that provide information supporting the answer are defined as Evidence. It can be observed that as the number of retrieved images increases, the total number of Evidence tokens also rises. However, the overall evidence density decreases rapidly. We compare the performance of our method with Qwen-7B (backbone) and OpenVLThinker (the strongest baseline) in terms of F1 score across different evidence densities. Our method consistently outperforms both baselines at

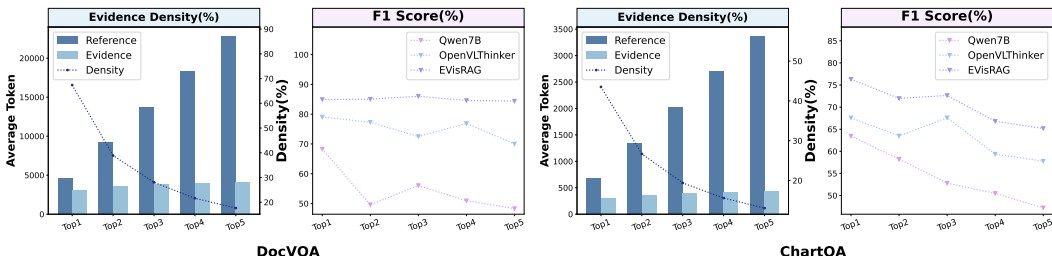

Figure 4: Performance comparison on different visual evidence density. Despite increasing noise with more retrieved images, EVisRAG maintains stable.

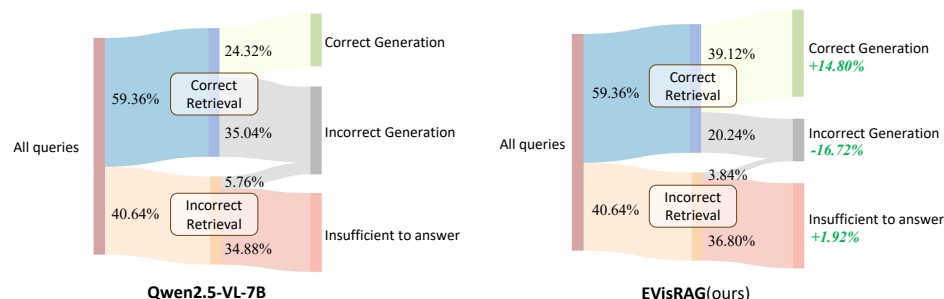

Figure 5: Model performance comparisons in different retrieval scenarios on ChartQA. Compared with the backbone, EVisRAG remains more faithful to the retrieved content in both correct and incorrect retrieval scenarios.

all density levels, demonstrating its superior generalizability in multi-image scenarios. Furthermore, on the DocVQA dataset, our approach maintains stable performance even as the evidence density decreases, highlighting its strong ability to resist hallucination effects.

## 5.5 IMPACT OF TRAINING ON MODEL PERFORMANCE

As shown in Figure 5, we examine the model's reasoning under varying degrees of contextual sufficiency to evaluate its balance between informativeness and hallucination. Before training, the baseline Qwen2.5-VL-7B displays a strong hallucination tendency even with correct retrieval—only 24.32% of queries yield correct generations, whereas 35.04% produce incorrect responses. Under incorrect retrieval, it predominantly abstains, reflecting a conservative strategy typical of smaller models. After training with our method, EVisRAG achieves a markedly better performance: the correct-generation rate in correctly retrieved contexts rises substantially, while a modest increase in abstention under incorrect retrieval is accompanied by a controlled reduction in incorrect generations. Overall, the trained model exhibits strengthened evidence-sensitive reasoning and reduced hallucination in underdetermined scenarios.

## 6 CONCLUSION

In this paper, we propose EVisRAG, a novel framework which enables vision-language models (VLMs) to observe–then–localize multi-image evidence during the thinking process to improve precise visual perception in multi-image scenarios with complex visual content. Specifically, EVisRAG introduces Reward-Scoped Group Relative Policy Optimization (RS-GRPO), which applies reward signals to specific token spans. This reward scope design improves the stability of long chain-of-thought (CoT) training and enables more accurate grounding of visual information throughout the reasoning process. Empirical results validate that EVisRAG effectively extracts key evidence from rich visual content, leading to improved reasoning accuracy. By equipping VLMs with fine-grained perceptual alignment across multiple images, EVisRAG marks a promising step toward more advanced and reliable visual retrieval-augmented generation (RAG) systems.

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

# A APPENDIX

## A.1 DATASETS

We evaluated five VQA benchmarks: InfoVQA, DocVQA, and SlideVQA were obtained from the VisRAG release (Yu et al., 2025) , ChartQA from its test split (Masry et al., 2022), and ViDoSeek from ViDoRAG (Wang et al., 2025a). Each dataset provides ground-truth answer image IDs. For each question, we retrieved the top-3 images using VisRAG-Ret as contexts. Following Joren et al. (2025), we labeled it sufficient if all ground-truth images were included, otherwise insufficient. The number of questions and sufficient context ratio in the dataset are shown in Table 3.

Table 3: Datasets used in our experiments.

| Name | #Questions | Description | Sufficient Context Ratio |
|---|---|---|---|
| ChartQA | 1250 | Visual and Logical Reasoning about Charts | 59.36% |
| InfoVQA | 718 | Question Answering on Infographic Images | 92.90% |
| DocVQA | 591 | Document Visual Question Answering | 83.59% |
| SlideVQA | 556 | Question Answering based on Multiple Slides | 89.93% |
| ViDoSeek | 1142 | Retrieval and Reasoning on Visually Rich Documents | 84.24% |

## A.2 DATA CONSTRUCTION OF GOLDEN REASONING TRAJECTORIES

For model training, we collected 30,000 samples from the ChartQA (Masry et al., 2022) and InfographicsVQA (Mathew et al., 2022) datasets, which were randomly divided into two subsets for SFT and GRPO in an 8:2 ratio. During the retrieval stage, VisRAG-Ret retrieves the top five candidate images for each query. While evaluation uses only the top three images as context, training leverages a variable number of retrieved images (top-1 to top-5) for data augmentation. Reasoning trajectories are constructed by generating candidate chains of thought with Qwen2.5-VL-72B and Qwen2.5-VL-7B (Bai et al., 2025), followed by a filtering process that retains only those trajectories yielding correct answers. This procedure generated 60,000 high-quality samples for SFT training, from which we extracted evidence as Ground Truth Evidence.

In the GRPO phase, we adopt a curriculum learning strategy following (An et al., 2025). Specifically, the SFT-trained model generates eight candidate completions for each sample, which are ranked according to their scores. Completions with perfect scores are excluded to mitigate overfitting. In addition, we incorporate 400 more challenging multi-hop examples from MMLongBench (Ma et al., 2024). The final GRPO training set consists of 4,000 carefully curated samples, organized to ensure a smooth progression from simple to complex instances, with a deliberate emphasis on more difficult cases to strengthen the model's reasoning robustness. The distributions of data difficulty before and after filtering are illustrated in Figure 6.

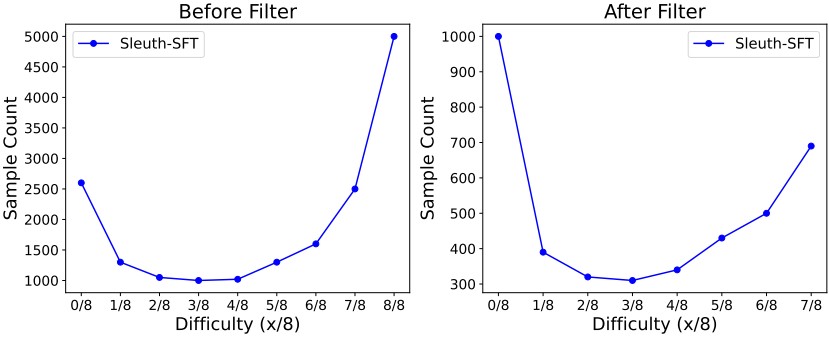

Figure 6: Data Difficulty Distribution of Before-Filtering and After-Filtering.

Table 4: Overall Performance of EvidenceCOT and Other MCOT.

| Methods | Single-hop | | | | | | Multi-hop | | | | Average | |
| | ChartQA | | InfoVQA | | DocVQA | | SlideVQA | | ViDoSeek | | | |
| | Acc | F1 | Acc | F1 | Acc | F1 | Acc | F1 | Acc | F1 | Acc | F1 |
|---|---|---|---|---|---|---|---|---|---|---|---|---|
| **MCOT** | | | | | | | | | | | | |
| COCOT | 49.52 | 46.71 | 31.34 | 26.96 | 40.95 | 32.66 | 35.25 | 29.52 | 33.80 | 33.93 | 38.17 | 33.96 |
| CCOT | 50.32 | 48.54 | 36.91 | 35.29 | 41.96 | 39.83 | 51.80 | 46.51 | 36.16 | 41.38 | 43.43 | 42.31 |
| DDCOT | 51.68 | 45.55 | 43.73 | 33.53 | 62.10 | 57.98 | 54.14 | 40.37 | 42.21 | 49.04 | 50.77 | 45.29 |
| Evidence-Guided Prompt(Ours) | **62.72** | **54.13** | **65.94** | **62.73** | **70.05** | **65.27** | **66.73** | **61.77** | **46.50** | **56.20** | **62.39** | **60.02** |

## A.3 MORE DETAILS ON THE FINE-GRAINED REWARD

In addition to stabilizing training to ensure accurate perceptual grounding and evidence-guided reasoning in VLMs, we further introduce five reward components, namely:

**Perception reward.** For text-only language models, using only answer accuracy as the reward signal together with GRPO training can elicit emergent "aha moments" and strengthen reasoning abilities (Guo et al., 2025). For VLMs, however, directly optimizing answer accuracy may improve reasoning while failing to improve perceptual accuracy (Liu et al., 2025). To optimize perceptual grounding and reasoning at the same time during training, we introduce a fine-grained perception reward:

$$R_{\text{perception}} = \frac{\sum_{i=1}^{n} r_i}{\sum_{i=1}^{n} (y_i \cdot k_{pos} + (1 - y_i) \cdot 1)}, r_i = \begin{cases} k_{pos} * f_1(e_i^{\text{pred}}, e_i^{\text{gold}}), & \text{if } y_i = 1 \\ \mathbb{I}(e_i^{\text{pred}} = \text{"no relevant information"}), & \text{if } y_i = 0 \end{cases} \quad (11)$$

The perception reward assesses whether the model extracts useful visual information. For each image, the evidence recorded by the model is compared with the corresponding gold evidence. For images that contain information relevant to the question, the reward is the F1 score between the predicted and gold evidence. For images that are irrelevant, the reward equals 1 if the model correctly indicates the absence of evidence and 0 otherwise. The final perception reward is the normalized sum of the image level rewards.

**Derivation reward.** We employ the F1-score between the predicted answer and the gold truth as the reasoning reward, where the gold truth is set to the fixed response "insufficient to answer" when the context is incomplete.

$$R_{\text{derivation}} = f_1(a^{\text{pred}}, a^{\text{gold}}), a^{\text{gold}} = \begin{cases} a^{\text{gold}}, & \text{if sufficient context} \\ \text{"insufficient to answer"}, & \text{if insufficient context} \end{cases} \quad (12)$$

where $a^{\text{pred}}$ denotes the model's predicted answer, $a^{\text{gold}}$ denotes the ground-truth answer, and $\text{Acc}_{\text{evi}}$ indicates whether the model's evidence predictions for all images are correct (assigned 1 if all are correct, and 0 otherwise).

**Format reward.** Beyond the accuracy-based reward, we also incorporate a format reward model that compels the model to follow our CoT design by sequentially performing observation, evidence recording, reasoning, and answering, with each stage encapsulated by its corresponding special tag (<observe>, <evidence>, <think>, <answer>).

$$R_{\text{format}}(a_i) = \begin{cases} 1, & \text{if the format of } a_i \text{ is correct} \\ 0, & \text{otherwise} \end{cases} \quad (13)$$

## A.4 IMPACT OF EVIDENCE-GUIDED REASONING

To evaluate the effectiveness of Evidence-Guided Reasoning, which explicitly encourages the VLM to first observe and record visual evidence before reasoning, we conducted two additional experiments. First, we compared our reasoning paradigm against three MCOT baselines, which also avoid additional training but attempt to enhance perception and reasoning by enforcing fixed prompting

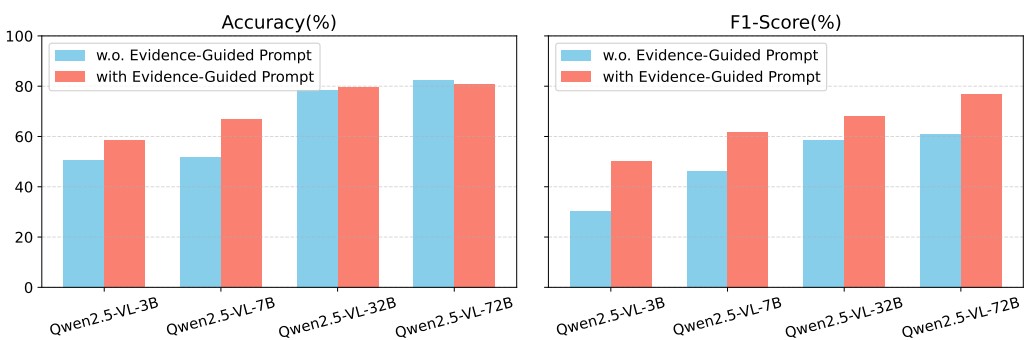

Figure 7: Performance comparison of Evidence-Guided Prompt Approach across different model sizes on the SlideVQA dataset.

patterns. As shown in Table 4, our approach consistently outperforms the baselines across five datasets. Although these MCOT strategies also prompt the model to improve perception by extensively describing image details, they tend to neglect the actual question. This often amplifies hallucinations by encouraging excessive descriptions. In contrast, our method records only question-relevant visual evidence, ensuring conciseness and enabling a more coherent and effective reasoning process. The three MCOT baselines are summarized as follows:

**DDCOT (Zheng et al., 2023).** A prompting strategy that decomposes complex questions into sub-questions and explicitly distinguishes between those requiring visual information and those that do not, thereby mitigating hallucinations and enhancing multimodal reasoning.

**CCOT (Mitra et al., 2024).** A prompting approach that leverages scene graphs as compact linguistic representations to enrich both image and task prompts, enabling LMMs to handle a wider range of vision-language tasks.

**COCOT (Zhang et al., 2024a).** A prompting strategy that improves the model's ability to capture fine-grained details in multi-image tasks by guiding it to explicitly identify similarities and differences between images.

Moreover, we further evaluated the generality of the Evidence-Guided Prompting approach across models of different scales. As illustrated in Figure 7, even without additional training, prompting the model to first record visual evidence and then reason upon it consistently improves both perception and reasoning across four different model sizes. This demonstrates the broad applicability and robustness of our proposed paradigm. prompt templates used by the EVisRAG are shown in Figure 11.

## A.5 MORE IMPLEMENTATION DETAILS

We acknowledge the contributions of LLaMA-Factory and EasyR1 (Yaowei et al., 2025) for releasing the training frameworks utilized in our SFT and GRPO experiments. We adopt Qwen2.5-VL-7B (Bai et al., 2025) as the backbone model for our proposed EVisRAG. EVisRAG is trained on 8× NVIDIA A100-80GB GPUs, with hyperparameters as shown in Tables 5 and 6.

## A.6 MORE IMPLEMENTATION DETAILS OF THE BASELINE METHODS

In this section, we provide comprehensive implementation details and prompt templates of the baseline methods evaluated in our study.

**General VLMs.** We assessed general vision-language models across different scales, namely Qwen2.5-VL-7B and Qwen2.5-VL-32B (Bai et al., 2025), as well as MiMo-VL-7B-RL (Xiaomi, 2025).

**VisRAG-Gen.** We additionally evaluate two generation strategies described in VisRAG (Yu et al., 2025).

Table 6: GRPO hyperparameters.

| Epoch | 4 |
|---|---|
| Rollout batch size | 32 |
| Global batch size | 32 |
| Max grad norm | 1.0 |
| Data type | bf16 |
| Learning rate | 1e-6 |
| Weight decay | 1e-2 |
| Warmup ratio | 0.0 |
| Rollout temperature | 1.2 |
| epsilon | 0.2 |
| epsilon_high | 0.28 |
| Image max pixels | 1568000 |

Table 5: SFT hyperparameters.

| Epoch | 1 |
|---|---|
| Data type | bf16 |
| Learning rate | 5e-7 |
| Global batch size | 32 |
| Scheduler | Cosine |
| Warmup ratio | 0.1 |
| Num train epochs | 1 |
| Image max pixels | 3920000 |

*Page Concatenation.* Page Concatenation forms a single composite image by horizontally concatenating the top-$k$ retrieved pages and feeds it to a single-image VLM. In our implementation, we adopt Qwen2.5-VL-7B (Bai et al., 2025) as the backbone VLM to ensure a fair comparison with other strong VRAG systems.

*Weighted Selection.* Weighted Selection instead generates an answer for each retrieved page independently and selects the final output based on the highest confidence, where the confidence weight combines the generation likelihood and the normalized retrieval score. For this method, we use the official implementation and pretrained MiniCPM-V-2 (Yao et al., 2024) model released by the authors. Together, these two variants represent the canonical generation pipelines of VisRAG and serve as competitive baselines in our evaluation.

**Vision-Language Reasoning Models (VLRMs).** We compare five fine-tuned VLRMs, all initialized from Qwen2.5-VL-7B-Instruct, each employing distinct strategies to enhance reasoning capabilities:

*Vision-R1-7B.* Vision-R1-7B (Zhan et al., 2025) introduces a reinforcement learning–based fine-tuning approach that incentivizes reasoning through vision-guided feedback. It circumvents the need for human-curated preference data by adopting a criterion-driven reward function.

*OpenVLThinker-7B.* OpenVLThinker-7B (Deng et al., 2025) follows an iterative two-stage training scheme, alternating between supervised fine-tuning (SFT) and reinforcement learning (RL). Starting from distilled reasoning competencies in text-only domains, the model progressively refines its reasoning by generating its own training data through RL and then using that data to further supervised fine-tune itself.

*MM-Eureka-7B.* MM-Eureka-7B (Meng et al., 2025) extends rule-based reinforcement learning (RL) to multimodal reasoning by incorporating new algorithms such as Online Filter, ADORA, and DAPO, which enhance reasoning efficiency and stability across multimodal tasks.

*Ocean-R1-7B.* Ocean-R1-7B (Lingfeng et al., 2025) builds upon a structured chain-of-thought evaluation framework that leverages knowledge graph exploration (e.g., OCEAN) to provide rich offline feedback, thereby aligning generated reasoning paths with factual knowledge.

*ThinkLite-VL-7B.* ThinkLite-VL-7B (Wang et al., 2025c) employs Monte Carlo Tree Search (MCTS)–guided sample selection to identify and train on genuinely challenging examples from a small dataset (11k samples), achieving state-of-the-art visual reasoning performance with high data efficiency.

**VRAGs (Visual Retrieval-Augmented Generation).** We further examine three advanced VRAG methods, all built upon the Qwen2.5-VL-7B-Instruct architecture:

*R1-Router.* R1-Router (Peng et al., 2025) employs a dynamic routing mechanism trained via Stepwise Group Relative Policy Optimization (Step-GRPO). R1-Router generates intermediate queries during the model's reasoning process and directs them selectively to the most appropriate knowledge base (e.g., text, image, table KB), harnessing the evolving reasoning state. This fine-grained

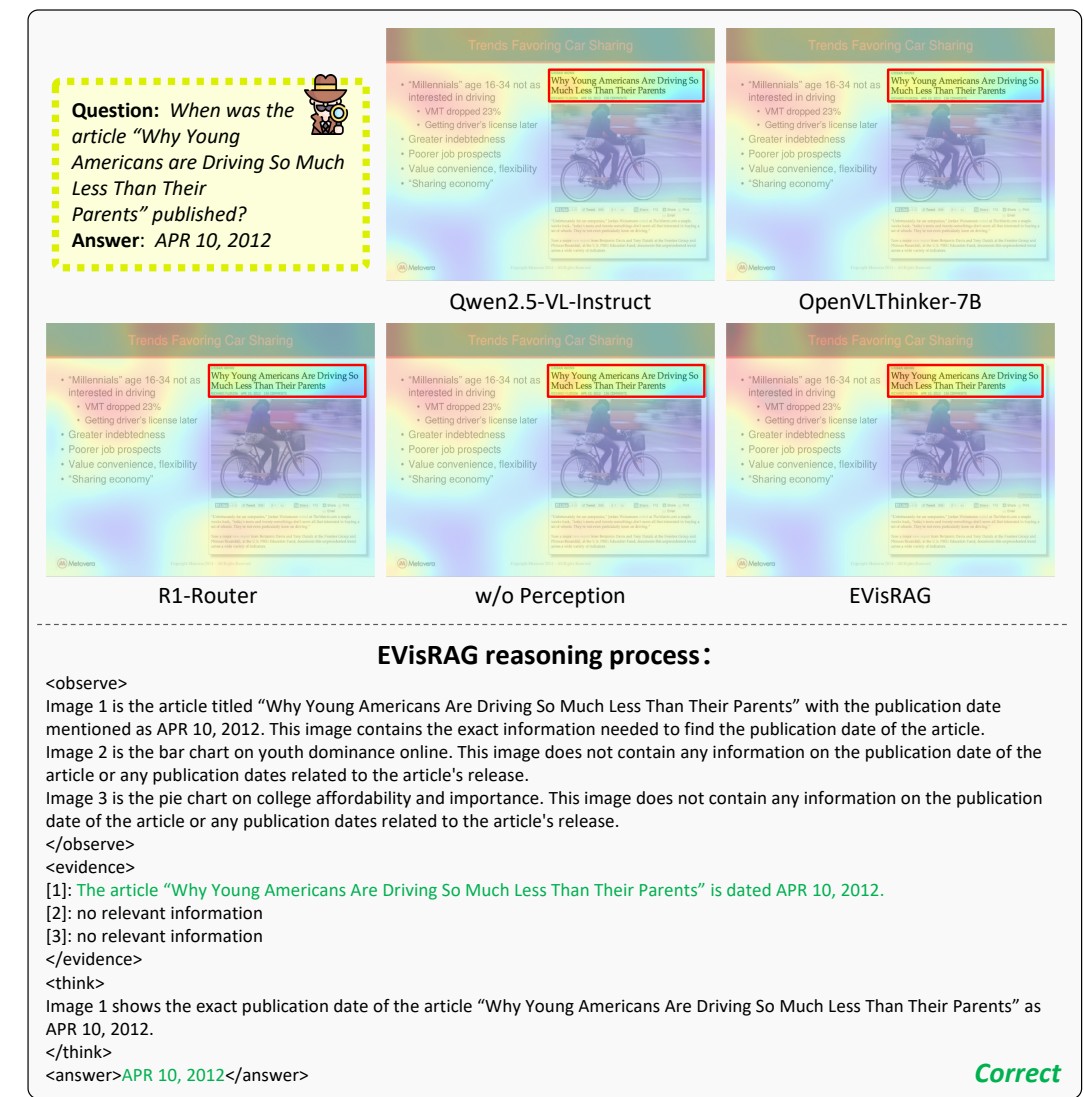

Figure 8: A Evidence Attention Case Study on SlideVQA.

routing capability enhances retrieval efficiency and reasoning precision by minimizing unnecessary retrievals while adaptively integrating external evidence.

*MMSearch-R1.* MMSearch-R1 (Wu et al., 2025) integrates multimodal search into the reasoning loop, employing cross-modal retrieval mechanisms to fetch contextually aligned information in both visual and textual forms.

*VRAG-RL.* VRAG-RL (Wang et al., 2025b) incorporates a reinforcement learning–based fine-tuning schema, enabling the model to progressively gather visual evidence from coarse to fine granularity and support multi-turn reasoning via an optimized retrieval-and-generation pipeline.

The prompt templates employed for each baseline are shown in Figure 12. For the three MCOT-based comparisons (DDCOT, CCOT, and COCOT), we adapted their original prompting strategies into an end-to-end chain-of-thought generation framework compatible with our setup. Their corresponding prompt templates are detailed in Figures 13, 14, and 15, respectively.

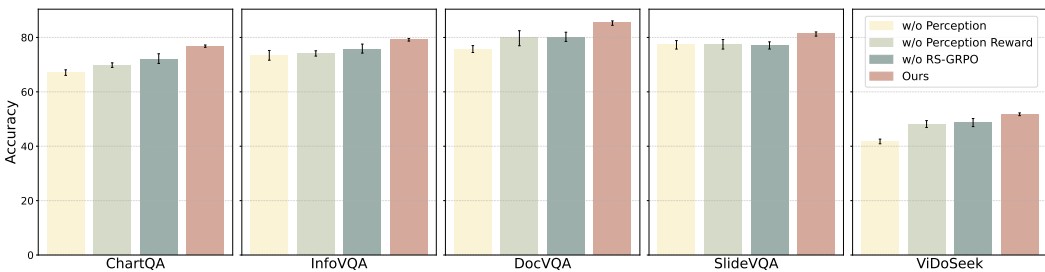

Figure 9: Ablation study(%): "w/o Perception" trains the model with a standard think-then-answer approach on the same data. "w/o Perception Reward" uses only answer correctness as the reward, omitting the additional Perception Reward. "w/o RS-GRPO" sums all rewards and applies them to every token, corresponding to the standard GRPO algorithm. Results are averaged over 5 runs with different random seeds, and error bars indicate 95% bootstrap confidence intervals.

## A.7 EXTENDED ABLATION WITH BOOTSTRAP CONFIDENCE INTERVALS

Figure 9 presents an extended visualization of the ablation study in Table 2, where we report *95% bootstrap confidence intervals* computed from five runs using different random seeds. The intervals are estimated via 10,000 bootstrap resamples for each method–dataset pair, providing a more reliable characterization of uncertainty compared to reporting only the mean and standard deviation.

Across all five benchmarks, the full EVisRAG model consistently achieves the highest accuracy, with its confidence intervals being well separated from those of the ablated variants in nearly all cases. This non–overlapping behavior indicates that the performance gains from Perception modeling, Perception Reward, and RS-GRPO are statistically significant rather than fluctuations due to random initialization. Moreover, the bootstrap intervals of our complete method are noticeably narrower, demonstrating more stable optimization dynamics. In contrast, removing any of the proposed components not only reduces accuracy but also increases variance, highlighting the necessity and robustness of each part of our reward design and training paradigm.

## A.8 MORE VISUAL ATTENTION CASES OF EVISRAG

We present in Figure 8 a qualitative comparison of attention alignment with question-relevant visual evidence. The query asks: When was the article "Why Young Americans are Driving So Much Less Than Their Parents" published? A human reader would first attend to the headline to verify topical relevance, then shift gaze to the metadata directly beneath it, where the publication date "APR 10, 2012" appears. As shown in the figure, EVisRAG places greater attention mass on these evidence regions than the baselines, and in its reasoning trace, explicitly observes and records the date "APR 10, 2012," yielding the correct answer. This case illustrates that EVisRAG enhances perception during reasoning by aligning attention with task-critical visual evidence.

## A.9 EVIDENCE-GUIDED REASONING OPTIMIZATION VIA RS-GRPO

To evaluate the effectiveness of our proposed Reward-Scoped Group Relative Policy Optimization (RS-GRPO), we compare its training dynamics with the standard GRPO baseline. As shown in Figure 10a, RS-GRPO consistently achieves higher answer rewards throughout training. While both methods exhibit fluctuations in early stages, RS-GRPO demonstrates a more stable upward trend and converges to a substantially higher reward level. This indicates that the fine-grained reward signals applied to in-scope tokens allow RS-GRPO to better align visual perception with reasoning, leading to more reliable improvements. Overall, these results confirm that RS-GRPO provides more effective optimization than GRPO, enabling EVisRAG to achieve superior reasoning quality.

## A.10 INFERENCE EFFICIENCY OF EVISRAG

The results in Figure 10b compare inference accuracy, latency, and output length on the ViDoSeek dataset across different approaches. Baseline models such as Qwen2.5-VL-7B-Instruct and Open-

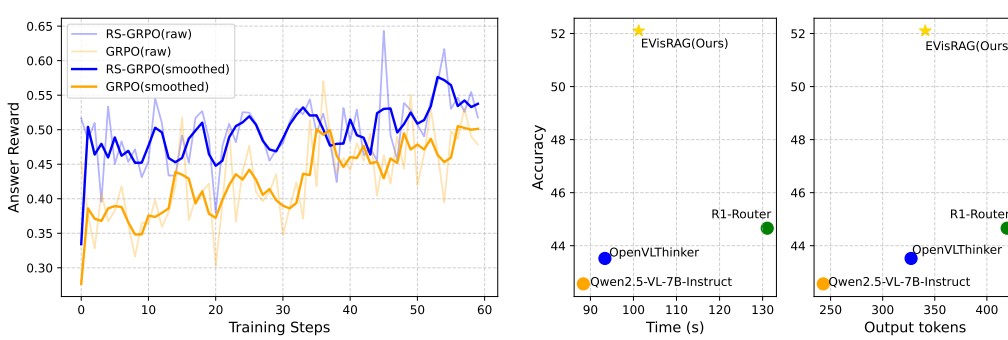

(a) Answer Reward Training Comparison      (b) Inference Efficiency Comparison of EVisRAG

Figure 10: Training and Inference Efficiency Comparison of EVisRAG

Table 7: Performance on natural-image QA.

|  | **In Distribution** (all images as input) | | | **Out of Distribution** (top-3 recall) | |
|---|---|---|---|---|---|
|  | 2 images | 3 images | 5 images | 10 images | 50 images |
| Qwen7b | 64.67 | 64.44 | 62.00 | 57.8 | 56.2 |
| EVisRAG(Ours) | **86.22** +21.55 | **85.33** +20.89 | **83.11** +21.11 | **71.6** +13.8 | **64.5** +8.3 |

VLThinker exhibit relatively low inference time (around 90–95 seconds) and short outputs (approximately 270–330 tokens), but their accuracy remains below 44%. R1-Router does not improve accuracy while incurring the highest computational cost: it requires the longest inference time (about 120 seconds) and produces the most verbose outputs (over 400 tokens) for a similar accuracy level. In contrast, our proposed EVisRAG, which adopts a single-step generation strategy, achieves a substantial accuracy gain of over 52% with only a modest increase in latency (around 100 seconds) and a moderate number of output tokens (about 300). These results show that EVisRAG delivers significantly better reasoning quality without sacrificing efficiency or introducing excessive verbosity, demonstrating its practicality for real-world applications.

### A.11    EXPERIMENTS ON NATURAL IMAGES

To further assess the generalizability of our method beyond document images, we additionally evaluate it on natural-image retrieval and reasoning under large-scale settings. We adopt the Visual Haystacks dataset (Wu et al., 2024) as both training and evaluation data. From the 2-image, 3-image, and 5-image configurations, we first select the same 100 questions for each setting, resulting in 300 training examples in total, and use the remaining 900 questions in each configuration as test data. In addition, we evaluate on the 10-image and 50-image configurations, using all 1,000 questions in each as test examples. For the 10- and 50-image settings, we employ `clip-vit-large-patch14-336`(Radford et al., 2021) to retrieve the top-3 most relevant images, which are then fed into the model.

We compare our trained model against the original *Qwen7B* model as the baseline. As shown in Table 7, our approach achieves more than 20% absolute accuracy improvement in the in-distribution settings (2, 3, and 5 images). In the out-of-distribution settings (10 and 50 images), even when relying on a relatively small CLIP model with imperfect retrieval quality, our method still yields on average more than 10% absolute improvement. These results demonstrate that our approach remains highly effective on natural-image tasks and is not limited to document-centric scenarios.

### A.12    ROBUSTNESS TO DIFFERENT RETRIEVERS

To examine whether our approach depends on a specific retrieval module, we further evaluate the trained model under multiple independent retrievers. Although our method is trained with VisRAG-

Table 8: Performance of EVisRAG and Qwen7b with different retrievers on ViDoSeek.

| | Sufficient Ratio | qwen7b-Acc | qwen7b-F1 | evisrag-Acc | evisrag-F1 |
|---|---|---|---|---|---|
| VisRAG-ret (Yu et al., 2025) | 84.24 | 42.56 | 42.48 | 52.10 +9.5 | 65.78 +23.3 |
| Colpali-v1.3 (Faysse et al., 2024b) | 84.33 | 42.23 | 40.95 | 50.79 +8.6 | 63.82 +22.9 |
| jina-embeddings-v4 (Günther et al., 2025) | 85.11 | 40.72 | 53.02 | 49.37 +8.7 | 64.01 +11.0 |

Ret as the retrieval component, at inference time we replace the retriever with two alternative models of different architectures and scales: Colpali-v1.3 and Jina-embeddings-v4. For each retriever, we obtain the top-3 relevant images and feed them into our QA model without any retraining or adaptation. The results in Table 8 demonstrate that our method yields consistent and substantial improvements across all retrievers, which confirm that our approach is retriever-agnostic.

### A.13 Case Studies of EVisRAG

In this section, we present three case studies illustrating EVisRAG's effectiveness: (i) single-hop QA, (ii) multi-hop QA, and (iii) alignment of attention with question-relevant visual evidence, each compared against strong baselines.

We begin with the single-hop case illustrated in Figure 16, drawn from the DocVQA dataset. In this example, the question asks for the name of the chemist listed in the document. EVisRAG correctly perceives and records the visual evidence, identifying that Richard W. Mann is annotated with the title chief chemist, and subsequently produces the correct answer, Richard W. Mann. In contrast, both OpenVLThinker and R1-Router misperceive the visual annotations during reasoning, mistakenly attributing the role of chemist to other individuals and thus generating incorrect answers.

We next analyze the multi-hop case in Figure 17 from the SlideVQA dataset. The question asks for the number of major languages in the country that governs mainland China and the largely self-governing territories of Hong Kong (since 1997) and Macau (since 1999). Answering requires integrating evidence from two slides: one identifies the country as China. The other enumerates China's major languages, including Mandarin, Yue (Cantonese), Wu (Shanghainese), Minbei (Fuzhou), Minnan (Hokkien–Taiwanese), Xiang, Gan, and Hakka, a total of eight. EVisRAG correctly records the provenance of each piece of evidence and produces the correct answer, demonstrating both reliable visual perception and cross-page reasoning. In contrast, OpenVLThinker and R1-Router fail: OpenVLThinker infers the correct subgoal but, having missed the second slide's list, predicts that no answer exists. R1-Router locates both slides but misperceives the list and counts seven instead of eight.

Finally, we examine a failure case from the ViDoSeek dataset (Figure 18) to highlight a remaining limitation of EVisRAG. The question asks which type of ISO standard for traditional Chinese medicine has the largest number of published standards in the regional report. EVisRAG correctly identifies the slide containing the relevant statistics and accurately parses fine-grained categories and counts from the bar chart, even recognizing that "Quality and safety of single herb (including seeds and seedlings)" attains the highest value among the subtypes. However, the question refers to the higher-level taxonomy on the slide, for which the correct answer is Chinese medicine. Because EVisRAG implicitly resolves the notion of "type" at the finer-grained subtype level, it bases its reasoning on the wrong semantic abstraction and consequently outputs an incorrect answer. This case shows that although our model can accurately perceive and organize detailed visual evidence, it may still produce errors due to a wrong understanding of the problem.

You are an AI Visual QA assistant. I will provide you with a question and several images. Please follow the four steps below:

**Step 1: Observe the Images**
First, analyze the question and consider what types of images may contain relevant information. Then, examine each image one by one, paying special attention to aspects related to the question. Identify whether each image contains any potentially relevant information.
Wrap your observations within *<observe></observe>* tags.

**Step 2: Record Evidences from Images**
After reviewing all images, record the evidence you find for each image within *<evidence></evidence>* tags.
If you are certain that an image contains no relevant information, record it as: [i]: no relevant information(where i denotes the index of the image).
If an image contains relevant evidence, record it as: [j]: [the evidence you find for the question](where j is the index of the image).

**Step 3: Reason Based on the Question and Evidences**
Based on the recorded evidences, reason about the answer to the question.
Include your step-by-step reasoning within *<think></think>* tags.

**Step 4: Answer the Question**
Provide your final answer based only on the evidences you found in the images.
Wrap your answer within *<answer></answer>* tags.
Avoid adding unnecessary contents in your final answer, like if the question is a yes/no question, simply answer "yes" or "no".
If none of the images contain sufficient information to answer the question, respond with *<answer>*insufficient to answer*</answer>.*

**Formatting Requirements:**
Use the exact tags *<observe>, <evidence>, <think>,* and *<answer>* for structured output.
It is possible that none, one, or several images contain relevant evidence.
If you find no evidence or few evidences, and insufficient to help you answer the question, follow the instruction above for insufficient information.

Question and images are provided below. Please follow the steps as instructed.
Question: {query}

Figure 11: The Prompt Template for EVisRAG(SFT&GRPO)

You are an AI assistant. I will provide a question and some images.

Put your reasoning process within *<think></think>.*
Please answer the questions based on the multiple pictures given to you, and put your your final answer in *<answer></answer>.*

Please try to remove irrelevant content in the final answer.
If you think there are no relevant information from the picture that can help you answer the question, answer *<answer>*insufficient to answer*</answer>* after your thinking.

Question: {query}

Figure 12: The Prompt Template for baselines.

You are an AI assistant. I will provide a query and some images. Follow these two steps:

**In the first step:**
Please think step-by-step about the preliminary knowledge to answer the question, deconstruct the question as completely as possible down to necessary sub-questions based on context, questions and options. Then with the aim of helping humans answer the original question, try to answer the sub-questions. The expected answering form is as follows:

**Sub-questions:**
1. <sub-question 1>
2. <sub-question 2>
...

**Sub-answers:**
1. <sub-answer 1> or 'Uncertain'
2. <sub-answer 2> or 'Uncertain'
...

For a question, assume that you do not have any information about the picture, but try to answer the sub-questions and prioritize whether your general knowledge can answer it, and then consider whether the context can help. If sub-questions can be answered, then answer in as short a sentence as possible. If sub-questions cannot be determined without information in images, please formulate corresponding sub-answer into "Uncertain".

**In the second step:**
Put your your final answer in *<answer></answer>* based on the scene graphs.
Please try to remove irrelevant content in the final answer. Like if the question is asking for yes or no, then only answer <answer>yes</answer> after your thinking.
If you think there are no relevant information from the picture that can help you answer the question, answer *<answer>*insufficient to answer*</answer>* after your thinking.

Question: {query}

Figure 13: The Prompt Template for DDCOT

You are an AI assistant. I will provide a query and some images. Follow these two steps:

**In the first step:**
For the provided images and its associated question, **generate a scene graph** for each images includes the following:

1. Objects that are relevant to answering the question
2. Object attributes that are relevant to answering the question
3. Object relationships that are relevant to answering the question

Then reason the answer of question based on scene graphs.
Put these process within *<think></think>.*

**In the second step:**
Put your your final answer in *<answer></answer>* based on the scene graphs.
Please try to remove irrelevant content in the final answer. Like if the question is asking for yes or no, then only answer <answer>yes</answer> after your thinking.
If you think there are no relevant information from the picture that can help you answer the question, answer *<answer>*insufficient to answer*</answer>* after your thinking.

Question: {query}

Figure 14: The Prompt Template for CCOT

You are an AI assistant. I will provide a query, and some images. Follow these two steps:

**In the first step:**
Find the **similarities and differences** of these images.
Output separately all the same points and all the differences you find.
Then reason the answer of question based on your findings.
Put these process within *<think></think>.*

**In the second step:**
Put your your final answer in *<answer></answer>.*
Please try to remove irrelevant content in the final answer. Like if the question is asking for yes or no, then only answer <answer>yes</answer> after your thinking.
If you think there are no relevant information from the picture that can help you answer the question, answer *<answer>*insufficient to answer*</answer>* after your thinking.

Question: {query}

Figure 15: The Prompt Template for COCOT

**Question:** *Who the only chemist is in the list?*
**Ground Truth:** *Richard W. Mann*

Evidence

------------------------------------------------------------------------

**EVisRAG reasoning process：**

<observe>
Image 1 is the cover of the book "Evaluating the Safety of Food Chemicals" by the National Academy of Sciences. It does not contain any information on any chemist's name, their profession, or any list of individuals and their occupations that could help identify the only chemist in the list. This image does not contain any relevant information.
Image 2 is the list of guests with their titles and affiliations. It does not contain any information on the profession of the individuals, their specific roles as chemists, or any list that confirms who the only chemist is in the list. No direct information on the profession of each individual as a chemist or any other professional title that confirms their role in chemistry.
Image 3 is the list of names with their titles and affiliations. It shows that Richard W. Mann is listed as the "Chief Chemist" from the Great Western Sugar Co. No other individual in the list is titled as a chemist or has a title that directly confirms their profession as a chemist. No other profession or title confirms any other individual as the only chemist in the list.
</observe>
<evidence>
[1]: no relevant information
[2]: no relevant information
[3]: Richard W. Mann is listed as the \"Chief Chemist\" from the Great Western Sugar Co.
</evidence>
<think>
Based on the evidence from Image 3, the only chemist in this list is Richard W. Mann.
</think>
<answer>Richard W. Mann</answer>                                    *Correct*

------------------------------------------------------------------------

**OpenVLThinker reasoning process：**

<think>
Okay, so I need to figure out who the only chemist is in the list provided. Let me start by looking at the images given. The first image is a book titled "Evaluating the Safety of Food Chemicals" by the National Academy of Sciences. That doesn't really help me with the list, but it might be related to the topic.
Moving on to the second image, it's a list of guests with their titles and affiliations. I'll go through each one to see if anyone is a chemist. The list includes people like Mr. Joseph F. Abbott, Miss Charlotte Adams, Mrs. Edith M. Barber, and so on. None of these names have "chemist" in their titles.
The third image is a list of names and their roles. It starts with Richard W. Mann, who is a MSG Plant in Johnstown, Colorado. Then there's Carl Peters, who is a Beet Grower Director. William Stewart is an Owner of KLMO. Ed Lehman is an Owner of Longmont Times Call. John Meyer is the President of Longmont National Bank. Robert Whyte is the President of St. Vrain Valley Bank. Reggie Batt is the President of First National Bank. Arthur G. Salberg is the President of First National Johnstown. Dr. R. K. Oldemeyer is a Director of Agricultural Research. Larry Flanders is a Company Attorney. Lloyd Dickens is a Farmer-Director of Production Credit. A. Frank Barnes is a Farmer. Harold Nesbitt is a Manager of A.S.C. Office. Art Stromquist is a Farmer.
Looking through this list, the only person with a role that could be related to chemistry is Dr. R. K. Oldemeyer, who is a Director of Agricultural Research. However, the question is specifically about a chemist, not an agricultural researcher. So, I need to see if anyone else has a chemist title.
After going through the list, I don't see any other chemists. The other roles are various business and agricultural positions. Therefore, the only chemist mentioned is Dr. R. K. Oldemeyer, but the question is asking for the only chemist in the list, which implies there's only one. Since the list doesn't mention any other chemists, the answer should be Dr. R. K. Oldemeyer.
</think>
The only chemist mentioned in the list is Dr. R. K. Oldemeyer, who is a Director of Agricultural Research. However, the question asks for the only chemist in the list, which implies there's only one. Since the list doesn't mention any other chemists, the answer should be Dr. R. K. Oldemeyer.
<answer>Dr. R. K. Oldemeyer</answer>                                *Incorrect*

------------------------------------------------------------------------

**R1-Router reasoning process：**

<think>
The question asks for the only chemist in a list. The information provided includes a list of individuals with their respective titles and affiliations. The list includes various professionals from different fields, including chemists. The relevant information can be found in the list of individuals, where titles such as "Chief Chemist" and "Chemist" are present.
</think>
<answer>
The only chemist in the list is Dr. Walter H. Eddy, who is listed as an American Public Health Association member. This information is based on the list of individuals provided, where Dr. Walter H. Eddy is listed with the title "American Public Health Association" next to his name.
</answer>                                                      *Incorrect*

Figure 16: A Case Study on DocVQA

**Question:** *How many major languages are there in the country that has control over mainland China and the largely self-governing territories of Hong Kong (since 1997) and Macau (since 1999)?*
**Ground Truth:** *8*

---

**EVisRAG reasoning process:**

<observe>
Image 1 is the Country Profile of the People's Republic of China (PRC). It mentions that the PRC has control over mainland China and the largely self-governing territories of Hong Kong (since 1997) and Macau (since 1999). This image contains the information needed to identify the country and its major languages.
Image 2 is the China Fact Sheet. It lists the major languages in China as Mandarin, Yue (Cantonese), Wu (Shanghainese), Minbei (Fuzhou), Minnan (Hokkien-Taiwanese), Xiang, Gan, and Hakka dialects. This image contains the exact information needed to find the number of major languages.
Image 3 is the map on Ethnolinguistic Groups in Afghanistan. This image does not contain any information on the major languages in the country that has control over mainland China and its territories, nor does it contain any information on the number of major languages in China or its control over the mentioned territories.
</observe>
<evidence>
[1]: The People's Republic of China (PRC) has control over mainland China and the largely self-governing territories of Hong Kong (since 1997) and Macau (since 1999).
[2]: The major languages in the People's Republic of China (PRC) are Mandarin, Yue (Cantonese), Wu (Shanghainese), Minbei (Fuzhou), Minnan (Hokkien-Taiwanese), Xiang, Gan, and Hakka dialects.
[3]: no relevant information
</evidence>
<think>
Image 2 lists the major languages in the People's Republic of China as Mandarin, Yue (Cantonese), Wu (Shanghainese), Minbei (Fuzhou), Minnan (Hokkien-Taiwanese), Xiang, Gan, and Hakka dialects. This image contains the exact information needed to find the number of major languages in the country that has control over mainland China and its territories.
</think>
<answer>8</answer>

*Correct*

---

**OpenVLThinker reasoning process:**

<think>
Okay, so I need to figure out how many major languages are spoken in the country that has control over mainland China and the self-governing territories of Hong Kong and Macau. Let me start by looking at the images provided.
First, I'll look at the Country Profile image. It mentions that the People's Republic of China (PRC) has control over mainland China and the self-governing territories of Hong Kong and Macau. That's the country we're looking for.
Next, I'll check the China Fact Sheet image. It lists the major languages spoken in China. The list includes Mandarin, Yue (Cantonese), Wu (Shanghainese), Minbei (Fuzhou), Minnan (Hokkien-Taiwanese), Xiang, Gan, and Hakka dialects. So, the question is asking about the number of major languages. The image mentions several dialects, but it doesn't specify how many are considered major. However, the image doesn't provide a clear cut-off point for what constitutes a major language versus a dialect.
Given that the image lists multiple dialects, it's possible that each one is considered a major language. However, the image doesn't provide a definitive number. It's also worth noting that the image doesn't mention any other languages besides these Chinese dialects. Therefore, based on the information provided, the number of major languages is not explicitly stated. The image suggests multiple dialects, but it doesn't give a specific number. So, the answer would be that there are multiple major languages, but the exact number is not provided.
</think>
The image mentions several dialects, but it doesn't specify how many are considered major languages. Therefore, the number of major languages is not explicitly stated.
<answer>insufficient to answer</answer>

*Incorrect*

---

**R1-Router reasoning process:**

<think>
The question asks for the number of major languages in the country that has control over mainland China and the largely self-governing territories of Hong Kong (since 1997) and Macau (since 1999). The images provided show information about China, including its official name, capital city, area, population, literacy rate, GDP per capita, major languages, currency, and time zone. The images do not contain information about Afghanistan, which is unrelated to the question.
The relevant information from the images is: 1. The People's Republic of China (PRC) has control over mainland China and the largely self-governing territories of Hong Kong (since 1997) and Macau (since 1999). 2. The major languages in China are Mandarin, Yue (Cantonese), Wu (Shanghainese), Minbei (Fuzhou), Minnan (Hokkien-Taiwanese), Xiang, Gan, and Hakka dialects. Based on the information provided, the answer to the question is:
</think>
<answer>
There are 7 major languages in the country that has control over mainland China and the largely self-governing territories of Hong Kong (since 1997) and Macau (since 1999). These languages are Mandarin, Yue (Cantonese), Wu (Shanghainese), Minbei (Fuzhou), Minnan (Hokkien-Taiwanese), Xiang, and Gan. Hakka dialects are also considered major languages in China.
</answer>

*Incorrect*

Figure 17: A Case Study on SlideVQA

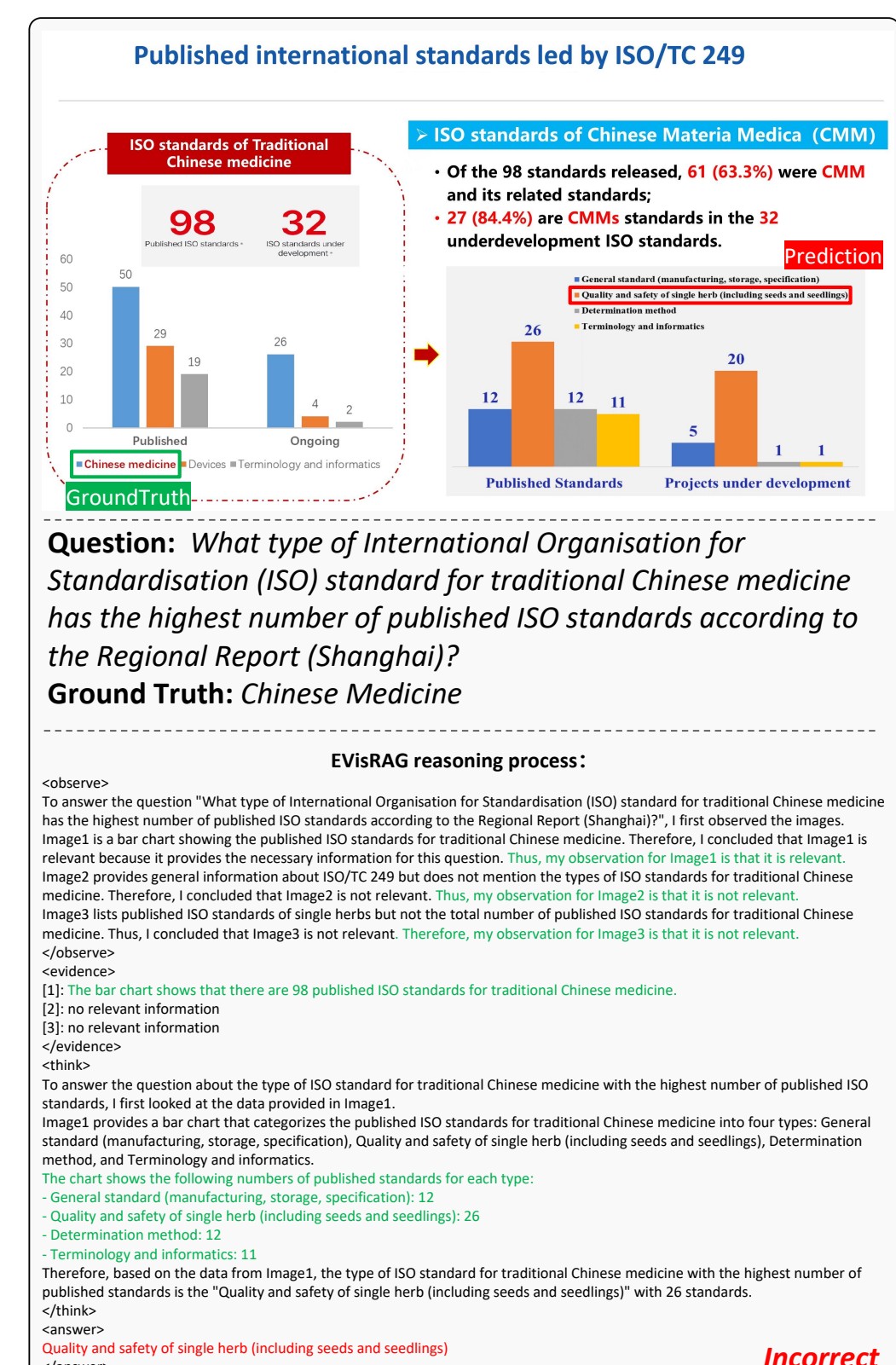

Question: *What type of International Organisation for Standardisation (ISO) standard for traditional Chinese medicine has the highest number of published ISO standards according to the Regional Report (Shanghai)?*

**Ground Truth:** *Chinese Medicine*

**EVisRAG reasoning process：**

<observe>
To answer the question "What type of International Organisation for Standardisation (ISO) standard for traditional Chinese medicine has the highest number of published ISO standards according to the Regional Report (Shanghai)?", I first observed the images. Image1 is a bar chart showing the published ISO standards for traditional Chinese medicine. Therefore, I concluded that Image1 is relevant because it provides the necessary information for this question. Thus, my observation for Image1 is that it is relevant. Image2 provides general information about ISO/TC 249 but does not mention the types of ISO standards for traditional Chinese medicine. Therefore, I concluded that Image2 is not relevant. Thus, my observation for Image2 is that it is not relevant. Image3 lists published ISO standards of single herbs but not the total number of published ISO standards for traditional Chinese medicine. Thus, I concluded that Image3 is not relevant. Therefore, my observation for Image3 is that it is not relevant.
</observe>
<evidence>
[1]: The bar chart shows that there are 98 published ISO standards for traditional Chinese medicine.
[2]: no relevant information
[3]: no relevant information
</evidence>
<think>
To answer the question about the type of ISO standard for traditional Chinese medicine with the highest number of published ISO standards, I first looked at the data provided in Image1.
Image1 provides a bar chart that categorizes the published ISO standards for traditional Chinese medicine into four types: General standard (manufacturing, storage, specification), Quality and safety of single herb (including seeds and seedlings), Determination method, and Terminology and informatics.
The chart shows the following numbers of published standards for each type:
- General standard (manufacturing, storage, specification): 12
- Quality and safety of single herb (including seeds and seedlings): 26
- Determination method: 12
- Terminology and informatics: 11
Therefore, based on the data from Image1, the type of ISO standard for traditional Chinese medicine with the highest number of published standards is the "Quality and safety of single herb (including seeds and seedlings)" with 26 standards.
</think>
<answer>
Quality and safety of single herb (including seeds and seedlings)
</answer>

*Incorrect*

Figure 18: A Failure Case Study on ViDoSeek

