# OpenReview forum: "Evidence-Guided Multi-Image Reasoning in Visual Retrieval-Augmented Generation"
_ICLR.cc/2026/Conference — ICLR 2026 Conference Withdrawn Submission_

### Official Review · Reviewer_nbuT · 2025-10-27

**Soundness:** 3
**Presentation:** 3
**Contribution:** 3
**Rating:** 4
**Confidence:** 4

**Summary:**

This paper identifies the limitation of poor cross-image evidence integration and weak grounding problems in current Visual Retrieval-Augmented Generation (VRAG) systems, and addresses this limitation by proposing EVisRAG, an end-to-end framework for multi-image evidence-guided reasoning. EVisRAG introduces explicit steps: (1) observe retrieved images, (2) record per-image evidence, (3) reason over aggregated evidence to generate answers. To optimize this framework, the authors present Reward-Scoped Group Relative Policy Optimization (RS-GRPO), which applies fine-grained rewards (perception, derivation, format) to task-specific token scopes to sharpen credit assignment. Empirical results across show EVisRAG outperforms baselines with impressive improvement on accuracy and F1 score.

**Strengths:**

1. EVisRAG’s step-by-step evidence recording solves the problem of implicit, ungrounded reasoning in VRAG systems.
2. Reward scoping addresses the blurring of credit assignment in mixed-reward training, leading to stable and superior performance.
3. Works across single/multi-hop tasks and diverse document types (charts, slides, docs), with prompt-based gains even for untrained models.

**Weaknesses:**

1. Efficiency: No detailed analysis of output token counts in the experiments. Explicit evidence steps may increase output length, but this trade-off is not quantified. Although the inference latency is provided in Sec. A.9, output token counts s a more appropriate metric for measuring inference costs than latency.
2. Failure Cases Exploration: Limited discussion of failure cases where EVisRAG misinterprets evidence or fails to handle the retrievals. How does EVisRAG perform with these situations?

**Questions:**

See Weaknesses.

---

> ### Author Response · Authors · 2025-11-19
>
> ### **W1: On the efficiency regarding output token numbers.**
> Thank you for this helpful suggestion. Following your feedback, we have expanded our efficiency analysis by explicitly reporting the output token lengths for all methods in addition to inference time and accuracy in Figure 10b. This provides a more complete view of practical efficiency. As shown in the updated results, baseline models such as Qwen2.5-VL-7B-Instruct and OpenVLThinker produce relatively short outputs (≈270–330 tokens) but remain below 44% accuracy, while R1-Router generates the longest outputs (over 400 tokens) and incurs the highest latency(\~120 s) without yielding accuracy gains. In contrast, EVisRAG achieves a substantially higher accuracy of over 52% while maintaining both moderate latency (\~100 s) and moderate output length (\~300 tokens). These results demonstrate that EVisRAG improves reasoning quality without increasing verbosity or computational overhead, confirming that our model remains both effective and efficient in real-world inference settings.
>
> ### **W2: On the Failure Cases Exploration.**
> Thank you for pointing out the need for a deeper analysis of failure cases. In the revised manuscript, we have added a failure-case study in Figure 18 that illustrates how EVisRAG behaves when misinterpreting the intent of the question despite accurately perceiving the visual evidence. In this case, EVisRAG correctly identifies the relevant slide, extracts all fine-grained numerical details from the bar chart, and pinpoints the subtype with the highest published standard count (“Quality and safety of single herb”). However, the model misaligns the semantics of “type” and resolves it at the fine-grained subtype level rather than the higher-level taxonomy required by the question (where the correct answer is “Chinese medicine”).
>
> This case demonstrates that EVisRAG’s errors stem not from failures in retrieval or visual grounding, both of which are handled correctly, but from misunderstanding the question’s intended abstraction level. Highlighting such cases in the paper helps clarify the model’s remaining limitations and provides a clearer direction for future improvements in semantic alignment and question-intent interpretation.

---

> > ### Comment · Reviewer_nbuT · 2025-11-27
> > **Reply to authors**
> >
> > Thank you for the rebuttal. I think the rebuttal addressed my concerns on W2. However, the concern regarding output token numbers still remains.
> >
> > One question concerns the inference procedure of EVisRAG: the output from each reasoning stage is fed as input to the next stage, that is, $r_{observe}, r_{evidence}, r_{reason}, r_{answer}$, with each stage producing a full response. The sum of the lengths of these stage-wise responses would be expected to be several times that of a vanilla single response. However, experimental results show that EVisRAG’s total output length (~300 tokens) is roughly the same as that of the base model (e.g., Qwen2.5-7B-VL-Instruct, ≈270 tokens). What factors contributed to this reduction in output length?

---

> > > ### Author Response · Authors · 2025-11-27
> > >
> > > Thank you for this question and for pointing out the apparent discrepancy. There is a misunderstanding in how the inference procedure of EVisRAG works.
> > >
> > > EVisRAG is an end-to-end model: during inference, the model generates $r_{observe}, r_{evidence}, r_{reason}, r_{answer}$
> > >  in a single pass, as one continuous response, rather than running four separate decoding stages and feeding the full output of each stage back as input. In other words, the model still produces only a single response sequence, just with structured segments corresponding to different reasoning stages, so the total number of generated tokens is of the same order as the base model.
> > >
> > > As shown in Figure 10(b), EVisRAG’s average output length is close to 350 tokens, while Qwen2.5-7B-VL-Instruct is below 250 tokens. We believe this additional overhead is acceptable in light of the performance gains brought by explicit multi-stage reasoning.

---

### Official Review · Reviewer_y76o · 2025-10-27

**Soundness:** 3
**Presentation:** 3
**Contribution:** 2
**Rating:** 4
**Confidence:** 4

**Summary:**

This paper presents a new visual retrieval-augmented generation framework, which introduces observation and evidence before think. And it introduces RS-GRPO to train the model. The approach demonstrates significant performance improvements across multiple VQA benchmarks and is well-structured in presentation. However, several concerns remain regarding its novelty and experimental design.

**Strengths:**

It achieves significant performance improvement.

The paper is clearly written and well-organized.

**Weaknesses:**

1. Although effectivenss, the motivation of introducing observation and evidence is unclear. The motivation of RS-GRPO is clear.
2. The introduction of observation and evidence appears to be an incremental improvement and not essential enough.
3. The selection of VRAG baselines is limited, and methods such as VisRAG are omitted. Moreover, the results in Table 2 show that a simple think-then-approach (without perception) already outperforms all baselines in Table 1.
4. The process for constructing SFT data with the newly introduced observation and evidence fields is not discussed.
6. Eq. 3-6 describe standard autoregressive generation processes and do not contribute meaningfully to the methodology. The rollout process is straightforward and does not require detailed exposition.

**Questions:**

See weaknesses.

---

> ### Author Response · Authors · 2025-11-19
>
> ### **W1 & W2: On the motivation and necessity of introducing observation and evidence.**
> We appreciate the reviewer’s question and would like to clarify that the introduction of explicit observation and evidencesteps is not an incremental heuristic, but a deliberate design that brings three concrete benefits.
>
> First, it decouples perception from reasoning, unlike standard think-then-answer paradigms where the model must simultaneously locate visual cues and perform multi-step reasoning. By prompting the model to “observe” before “thinking,” we allow it to form a grounded, structured perception of relevant visual content, which significantly improves reasoning quality even without any additional training, **as shown in Table 4, where simply adding observation prompts yields large gains across model sizes**.
>
> Second, this design substantially improves interpretability and trustworthiness: the model is encouraged to mimic how humans answer visual questions, first identifying question-relevant visual evidence, then reasoning over it, producing outputs that explicitly show what was perceived and why the final answer follows.
>
> Third, the structured four-step format (Observe, Evidence, Reason, Answer) enables finer-grained optimization during RS-GRPO training: each stage is marked with special tokens, allowing different reward signals to be aligned with the appropriate step. This step-level supervision leads to more stable training and clearer improvements, as demonstrated in ablation study (Table 2), where removing perception or perception reward consistently degrades performance with statistically significant drops. Overall, the observation–evidence design is essential to unlocking the benefits of RS-GRPO, enhancing interpretability, and materially improving performance, rather than being a minor modification.
>
> ### **W3: On the selection of VRAG baselines and the performance of the simple think-then-answer approach.**
> We thank the reviewer for pointing this out. In the revised version, we have added VisRAG-Gen to Table 1 now. Regarding the concern that the simple think-then-answer model appears strong, Table 1 and Table 2 show that this variant is only comparable to the best existing baselines (e.g., Qwen2.5-VL-32B or OpenVLThinker), but remains far below EVisRAG, with gaps of up to ~10% accuracy on average. Even EVisRAG-3B consistently outperforms the simple think-then-answer baseline across all datasets. These results highlight that although the vanilla reasoning pipeline is reasonably strong, our proposed perception-and-evidence design and RS-GRPO optimization together yield substantial and consistent improvements, demonstrating that EVisRAG provides benefits beyond what existing RAG baselines and simple prompting strategies can achieve.
>
> ### **W4: On the process for constructing SFT data.**
> We clarify that the SFT data and the GRPO data are constructed from the same source process using the same data pipeline described in Section 4 Datasets and Appendix A.2. Specifically, all reasoning trajectories, including the newly introduced observation and evidence steps, are generated by the Qwen2.5-VL-72B model, which we use to produce full structured chains of thought. If you have specific aspects of the data construction you would like us to elaborate on, we would be happy to clarify further.
>
> ### **W5: On the comment that Eq. 3–6 cover standard autoregressive generation and may not be necessary.**
> We appreciate the reviewer’s feedback. Our intention was to provide a self-contained description of the rollout process for clarity. We will revise the final version by simplifying redundant equations and focusing the exposition on the novel components of our method.

---

> > ### Comment · Reviewer_y76o · 2025-11-24
> > **Reply to authors**
> >
> > Thank you for the rebuttal. However, it still doesn't resolve my concerns, and I will maintain my score for the following reasons:
> >
> > (1) The motivation is still not clear enough. The authors emphasized the benefits of evidence in the rebuttal, but is it really that bad to not include evidence? I suggest adding more analysis; Figure 1, with only one case, is not convincing enough. Furthermore, the ablation experiments in Table 2 show that the performance is similar without evidence. In addition, adding evidence is not novel enough, because evidence-based reasoning is already prevalent. Applying it to a different field makes little difference.
> >
> > (2) The author included a comparison with VisRAG, but why not compare the more powerful performance in Tables 2/3 of VisRAG? In fact, this performance is similar to that reported in these tables.
> >
> > (3) "Moreover, the results in Table 2 show that a simple think-then-approach (without perception) already outperforms all baselines in Table 1." The author ignored my question. If all results in the ablation experiments exceeded the baselines in Table 1, what caused such a significant performance improvement over the baselines? This makes the contribution of this paper unclear.
> >
> > (4) The data construction process needs to be included in the main text, as constructing evidence data is a crucial part of this paper. I suggest adding a figure. Replacing Equations 3-6 with a description of the data construction process would be more meaningful.

---

> > > ### Author Response · Authors · 2025-11-25
> > >
> > > ### **Response to Q1.**
> > >
> > > Thank you very much for your detailed comments on the motivation and the role of evidence.
> > >
> > > **Regarding “is it really that bad to not include evidence?”:**
> > >
> > > Our ablation in Table 2 indeed shows that the variant without explicit evidence already improves over the baseline, which we fully acknowledge. At the same time, the variant with evidence consistently brings a larger gain: across the main benchmarks, including evidence yields more than 7 percentage points absolute improvement over the no-evidence variant. We view this gap as non-trivial in the context of multimodal document QA, and it suggests that evidence does not merely add complexity but actually raises the performance ceiling.
> > >
> > > In addition, Table 4 shows that even in a pure prompt-based setting (without RL training), explicitly including evidence in the prompt already improves model performance. This indicates that evidence is beneficial both as an intermediate training signal in RS-GRPO and as an explicit, human-interpretable support for the final answer.
> > >
> > > If you feel that this still does not fully demonstrate the importance of evidence, we would be very grateful for your guidance on what kind of additional analysis or experiments you would find most convincing to support our motivation.
> > >
> > > **On your suggestion to add more analysis, and the concern that Figure 1 with a single case is not sufficiently convincing:**
> > >
> > > In the appendix (Figures 8, 16, and 17), we have already included three additional representative cases that illustrate how the selected evidence affects reasoning, and how missing or incorrect evidence leads to failures. In a further revision, we are happy to add more cases if you think particular types of examples would be especially helpful. If you have specific preferences for what kinds of qualitative cases you would like to see, we would greatly appreciate your suggestions and can tailor the additional examples accordingly.
> > >
> > > **Concerning the point that “the ablation experiments in Table 2 show that the performance is similar without evidence”:**
> > >
> > > As mentioned above, our method with evidence improves over the no-evidence variant by more than **7 absolute points** on the main benchmarks. From our perspective, this difference is quite substantial in this domain.
> > >
> > > **Regarding the comment that “adding evidence is not novel enough, because evidence-based reasoning is already prevalent”:**
> > >
> > > In our literature review on document image understanding and multi-image RAG, we did not find prior work that explicitly introduces and optimizes intermediate evidence in the manner proposed in this paper. If there are specific methods that you believe already realize essentially the same idea, we would be very grateful if you could point us to the corresponding references. We will carefully study them and update the related work and discussion to clearly position our contribution with respect to these methods.
> > >
> > > Moreover, in the document image domain, first gathering visual evidence from multiple pages and then reasoning over it closely mirrors human reading behavior, and the resulting evidence chains substantially improve interpretability in addition to performance.
> > >
> > > **Finally, regarding the remark that applying it to a different field makes little difference:**
> > >
> > > Our view is that in the document image scenario, introducing explicit, region-grounded evidence is especially natural and valuable: it aligns with how humans inspect different parts of a page (and across pages) to answer questions, and it makes the model’s decision process more transparent. The experimental results show that this design yields substantial performance improvements on multi-image document QA tasks as you said in the strength-1, while also providing interpretable intermediate evidence steps. Given both the quantitative gains and the interpretability benefits in this specific multi-image RAG setting, we believe the motivation and contribution are meaningful for this field.

---

> > > ### Author Response · Authors · 2025-11-25
> > >
> > > ### **Response to Q2.**
> > >
> > > Thank you for pointing this out and for encouraging a clearer comparison with the results reported in VisRAG.
> > >
> > > First, Table 2 in VisRAG mainly reports **retrieval performance**, whereas in this work we focus on evaluating the end-to-end generation quality of the RAG system. Our main results therefore use answer accuracy metrics on downstream QA benchmarks, rather than standalone retrieval scores.
> > >
> > > Second, regarding Table 3 in VisRAG, we now make the comparison to their strongest reported settings explicit. Below we summarize the numbers reported in VisRAG Table 3 (using both MiniCPM-V and GPT-4o as the backbone) and compare them with EVisRAG under the same Top-3 retrieval setting:
> > > |Method|ChartQA|DocVQA|InfoVQA|SlideVQA|
> > > |-|-|-|-|-|
> > > |VisRAG(Top-3) on MiniCPM-V|42.76|66.05|45.45|49.45|
> > > |VisRAG(Oracle) on MiniCPM-V|65.74|82.01|55.57|56.83|
> > > |VisRAG(Top-3) on GPT-4o|46.10|64.93|56.45|56.34|
> > > |VisRAG(Oracle) on GPT-4o|65.88|78.45|65.10|62.74|
> > > |EVisRAG(Top-3)|**76.80**|**85.45**|**79.39**|**81.29**|
> > >
> > > As shown in the table, **EVisRAG with Top-3 retrieval not only outperforms VisRAG under the same Top-3 setting, but also surpasses the Oracle variants of VisRAG that use golden images (both on MiniCPM-V and GPT-4o backbones) across all four benchmarks.** This indicates that our RS-GRPO–based evidence reasoning and generation pipeline brings substantial additional gains beyond what can be achieved by simply pairing VisRAG’s retriever with a strong backbone.
> > >
> > > ### **Respond to Q3.**
> > > We believe a major factor behind the performance gap is that all variants in Table 2 are trained with our RL framework in the document-image RAG setting, whereas the baselines in Table 1 are standard off-the-shelf methods without such task-specific RL adaptation.
> > >
> > > On this common RL-trained backbone, we then gradually add our proposed components, and the performance increases step by step, which indicates that each module makes a non-trivial contribution rather than the gain coming from a single factor.
> > >
> > > ### **Respond to Q4.**
> > > Thank you for highlighting the importance of the evidence data construction process. We agree that this component is crucial for understanding our method.
> > >
> > > We kept Equations (3)–(6) in the main text because they help readers clearly follow the overall reasoning pipeline, in a way similar to how Search-o1[1] and related work formally describe their reasoning steps. At the same time, we fully agree that a more explicit description of the data construction process would further improve clarity.
> > >
> > > Due to the 9-page limit during the rebuttal phase, we were not able to move Section A.2 into the main body in the current version. However, in the camera-ready version we plan to integrate the key details of the data construction process from Appendix A.2 into the main text, and add a new figure that visually illustrates how the evidence data is constructed. We believe this will make the role of evidence construction in our framework much clearer.
> > >
> > > [1] Li, et al. "Search-o1: Agentic Search-Enhanced Large Reasoning Models", 2025.

---

> ### Comment · Reviewer_cc7b · 2025-11-26
>
> Joining this discussion as I found that my concerns are really similar to those raised by Reviewer y76o. I'm okay with most of the authors' replies. One thing I wanna point out is that: I think the response to Q2 is not fair cuz EVisRAG is a reasoning model while other baselines (MiniCPM-V and GPT-4o) are instructed models. Please correct me if I was wrong.

---

> ### Author Response · Authors · 2025-11-27
>
> **Clarification on the comparison with VisRAG.**
> Regarding the concern that EVisRAG is a reasoning model while MiniCPM-V and GPT-4o are instruction-tuned models, we would like to clarify three points.
>
> 1. **Choice of MiniCPM-V and GPT-4o.**
>
> Our response to Q2 is based on the requirements of reviewer y76o. Specifically, we report the MiniCPM-V and GPT-4o results that VisRAG itself uses in Table 3 as their strongest configurations, rather than selecting alternative or weaker settings on our own. In other words, the choice of backbones and configurations in this comparison is determined by the original VisRAG paper, not by us.
>
> 2. **Instruction-tuned vs. reasoning-tuned models.**
>
> The instruction-tuned model is not inherently weaker than the reasoning-tuned model itself. In fact, GPT-4o is a very capable model with strong multi-step reasoning abilities. Comparing EVisRAG, which is built upon a reasoning model with about 7B parameters, to GPT-4o does not imply that EVisRAG has an unfair advantage in terms of the original model's capacity or general reasoning ability.
>
> 3. **Consistency with other reasoning-style baselines and our own backbone.**
>
> In addition, Table 1 in our paper already includes baselines that are also reasoning-style models (such as VLRM-type methods), so EVisRAG is compared not only against purely instructed systems. Moreover, the base model we train in EVisRAG is itself an instruction-tuned model, **Qwen2.5-VL-7B-Instruct**. The **reasoning behavior comes from our training paradigm** (RS-GRPO with evidence), rather than from using a fundamentally different type of backbone. In this sense, the comparison is primarily about the training and evidence framework, not about giving EVisRAG a stronger pre-trained model.

---

> ### Comment · Reviewer_cc7b · 2025-11-27
>
> Okay sorry I missed the context (though I understand it's requested by the reviewers, I still personally think that running that comparison is kinda weird given that not all of them are reasoning models); also, I have updated the score in my section as the reviews address my concerns.

---

### Official Review · Reviewer_cc7b · 2025-10-30

**Soundness:** 2
**Presentation:** 3
**Contribution:** 2
**Rating:** 4
**Confidence:** 4

**Summary:**

The authors present a single model, EVisRAG, that does a visual-RAG-like operations in the chain-of-thought to yield better results compared to the visual reasoning models. Specifically, when given a question and a set of images, the model will first go through all i mages, determine if they're relevant or not, do reasoning, and answer the question eventually. The authors proposed RS-GRPO, a variant of GRPO, to provide rewards at different part (such as one reward for reasoning, another for relevance prediction) during the RL training. The proposed pipeline improves the performance compared to the baseline on multiple benchmarks.

**Strengths:**

1. The paper is clear and easy to follow.
2. The proposed method is effective on multiple benchmarks and the analyses in "insufficient to answer" is interesting.

**Weaknesses:**

- While the paper is technically sound, the gain of the perception reward and the RS-GRPO seems to be small in both in-distrubiton and OOD settings (Table 2). The difference of attention maps in Figure 3 seems to be small as well. As the RL part is the key technical contribution of the paper, it would be great if the author can provide the error bar (bootstrapped average with confidence interval) to show that the proposed method is actually useful or not. Also, it would be great if we can try natural images (beyond documents only) with large-scale settings [1, 2].

- The reviewer believes that the system is slightly different from the actual two-stage RAG setting. What's the advantage of training one large model as a whole compared to having one retriever and one generator. The most efficient setting would be a light-weight retriever to get top-K images, let's say CLIP or other post-trained variants [1]. In a real-world setting with large-scale documents, going through each image one by one in a reasoning model can be time-consuming and easily runs out of the context window. Another baseline that worth comparing would be calling a VLM twice with the first time doing retrieval and the second one doing generation based on retrieved images.


[1] Wu, Tsung-Han, et al. "Visual haystacks: A vision-centric needle-in-a-haystack benchmark." ICLR 2025.

[2] Chen, Jun, et al. "Document haystacks: Vision-language reasoning over piles of 1000+ documents." CVPR 2025.

**Questions:**

please read the weakness part.

---

> ### Author Response · Authors · 2025-11-19
>
> ### **W1-1. On the perceived small gain of the perception reward and RS-GRPO.**
> Thank you for raising this concern. To more clearly demonstrate the effectiveness and stability of each component, we have strengthened the ablation protocol in Sec. 5.2: each approach is now run 5 times with different random seeds, we report the mean accuracy across runs, and we add 95% bootstrap confidence intervals (error bars) in the updated Fig. 9. Under this setting, the contributions of Perception, Perception Reward, and RS-GRPO become much clearer: our full model consistently outperforms all ablated variants on all datasets, and the confidence intervals of EVisRAG are largely non-overlapping with those of the ablations, indicating that the gains are stable and statistically meaningful rather than measurement noise.
>
>
> ### **W1-2: On comparisons with natural images.**
> In response to the reviewer’s suggestion, we have added a new section “Experiments on Natural Images” evaluating EVisRAG on the Visual Haystacks natural-image QA benchmark under large-scale multi-image settings (Section A.11). As shown in Table 7, our method achieves more than +20% absolute accuracy improvement over Qwen-7B in the in-distribution 2/3/5-image settings, and still obtains around +8–14% absolute gains in the out-of-distribution 10/50-image settings, even when relying on a relatively small CLIP retriever. These results demonstrate that EVisRAG generalizes effectively to natural-image reasoning and is not limited to document-centric scenarios.
>
> ### **W2: On the claim that our system is “slightly different” from a standard two-stage RAG setting**
> We would like to clarify that EVisRAG in fact follows a canonical two-stage RAG pipeline, but our contributions focus on how to use the retrieved images on the generation side rather than on changing the retrieval architecture. As illustrated in Fig. 1, we first use an external retriever to obtain the top-K images (K=3 in our main experiments, which can be instantiated by VisRAG-Ret, Colpali, Clip, etc.), and then feed the question together with these retrieved images once into EVisRAG (right side) for end-to-end reasoning; the model does not scan all images one by one inside the generator, so the context window and latency are governed by K rather than by the total number of images in the corpus. This design preserves the efficiency benefits of a light-weight retriever. Moreover, as discussed in Section A.12 and shown in Table 8, replacing the original retriever VisRAG-Ret with Colpali-v1.3 or Jina-embeddings-v4 at inference time, without any retraining, still yields consistent and substantial gains over the corresponding baselines, indicating that EVisRAG is retriever-agnostic rather than tied to a specific retrieval model.

---

> > ### Comment · Reviewer_cc7b · 2025-11-20
> > **Reviewer's Response**
> >
> > Thanks for the rebuttal. I think the revised PDF and the rebuttal content addressed most of my concerns. I'm leaning towards increasing the score because the first weakness is not there after the revision. (I did not want to promises it as I will also read other reviewers' point later as well)
> >
> > For the second weaknesses, I now understand you are considering a 3-image reasoning scenarios and there will be a retriever there in advance. I would suggest adding the problem setting to the introduction and/or the beginning of section 3.1 in the next revision such that the reader knows that you only focus on the generation part (otherwise the one-line text in Figure 1 is really easy to miss...)
> >
> > One final question: does the RS-GRPO paradigm also works within a single image (but different image) or the authors think it would benefit more for multi-image tasks and single-image tasks are all saturated?
> >
> > Thanks.

---

> ### Author Response · Authors · 2025-11-21
>
> Thank you very much for your follow-up and for considering increasing your score.
>
> We have further revised Section 3.1 (L150–L155 in the updated version) to make the problem setting explicit. In particular, we now clearly state that EVisRAG first performs information retrieval over top-k candidate images, and that our method focuses on the subsequent generation module. We hope this makes it much easier for readers to understand the setting.
>
> Regarding your question about whether the RS-GRPO paradigm also works within a single image, in our understanding, the single-image case corresponds to using only the top-1 retrieved image. In this setting, RS-GRPO still works well. As shown in Figure 4, our method continues to outperform other baselines even under top-1 retrieval.
>
> For the different image setting, we understand this as referring to natural image scenarios. As shown in Table 7 and Appendix A.11, RS-GRPO also brings consistent and often substantial improvements over baselines on natural image tasks.
>
> By the way, in the ICLR review form, a score of 8 corresponds to “Accept (poster)”. We would be very grateful for a corresponding update of the score if you feel that our revisions have adequately addressed your concerns. Thank you again for your very helpful comments, and we are happy to clarify any further questions you may have about the paper.

---

### Author Response · Authors · 2025-11-19

We sincerely thank all reviewers for your thoughtful and constructive comments. Following your suggestions, we have revised and enriched several tables, figures and descriptions in the paper, and these updates have been reflected in the current version of the PDF.

---

### Author Response · Authors · 2025-11-24
**We kindly invite you to review our latest responses in ICLR2026**

Dear Reviewers,

We sincerely thank you for your time and effort in reviewing our paper. Your detailed feedback is invaluable to us.

We understand you may have a packed schedule and other commitments, but we are looking forward to hearing from you soon. Your suggestions are crucial for us to further enhance our work. Could you please share your updated thoughts and advice with us?

We commit to actively responding to your suggestions to ensure our work meets the highest standards. Many thanks again for your time and favourable consideration.

Best regards,

The Authors of Submission817

---

### Note · Authors · 2026-01-04

I have read and agree with the venue's withdrawal policy on behalf of myself and my co-authors.